# Optimal Operation of a Benchmark Simulation Model for Sewer Networks Using a Qualitative Distributed Model Predictive Control Algorithm

**Antonio Cembellín** [1,*] , **Mario Francisco** [1] **and Pastora Vega** [2]

1 Computing and Automation Department, Higher Technical School of Industrial Engineering, University of Salamanca, 37700 Béjar, Spain; mfs@usal.es
2 Computing and Automation Department, Faculty of Science, University of Salamanca, 37008 Salamanca, Spain; pvega@usal.es
* Correspondence: cembe@usal.es; Tel.: +34-923-4080-800 (ext. 2237)

**Abstract:** This article presents a distributed model predictive control algorithm including fuzzy negotiation among subsystems and a dynamic setpoint generation method, applied to a simulated sewerage network. The methodology considers WWTP as an additional objective of control. To improve the performance of a DMPC using a hydraulic model for prediction, a more detailed model has been considered including suspended solids concentration (TSS). The results obtained with the proposed methodology have been validated on a benchmark simulation model for sewer systems developed to test and compare methodologies, showing good performance.

**Keywords:** distributed model predictive control (DMPC); sewer network; fuzzy logic

## 1. Introduction

One of the most important resources for humans on earth is water, which becomes easily polluted due to human activities. An adequate wastewater treatment may clean the water to be used again, making this cycle sustainable. Urban drainage systems (UDS) collect and transport both wastewater resulting from human activity and rainfall to treatment facilities avoiding discharges into the environment, forming a combined urban drainage system (CUDS). These systems mainly consist of storage tanks and connecting pipes, constituting a sewer network. After heavy rain events, the wastewater obtained can overflow the network and pollute the environment.

Obtaining mathematical models of the process that enable the design of different control algorithms has been the main research in this field. The operating objectives are essentially to avoid losses of wastewater from the network due to overflows in the storage reservoirs and at the inlet to the WWTP and to maximize this inlet flow, decreasing the economic cost of operation [1,2]. Moreover, reducing the pollutant mass escaping from the network may be a prior objective, so it is needed to consider the concentration of pollutants in the overflowed water [3,4]. Another problem is the "first flush" when abundant precipitation falls after a dry season, provoking a large growth in the concentration of contaminants in the sewer system [5].

Simulation models must accurately describe the processes that occur in a sewerage system (hydraulic and pollutant mass transport). They must be simple enough to be used for control. The features of the wastewater caught depending on the area (urban or industrial wastewaters with variable concentrations of contaminants) and the rainwater going into the network, as well as the specific period of the year, are considered in these models. In consequence, typical patterns that attempt to represent different situations have been developed [6]. Some of them are benchmarks for testing control systems of the sewage, such as [5], and simpler ones can be obtained to be used for advanced control

design algorithms. In general, control strategies for UDS [7] can be off-line controllers, which use static rules, and on-line controllers that use real-time control actions [8–10]. Within the first group, RBC (rule-based control) algorithms have been used in [11,12], with their main problem being the number of rules when the scale of the system grows. Control systems based on fuzzy logic control (FLC), combining simple rules with an expert system, are applied in [13–16]. Within the second group of algorithms, the LQR controller has been used in [17] to reduce the overflows in the network by using all the available storage volume and emptying the system immediately, if possible. Additionally, genetic algorithms have been used in water quality networks in [3] and to optimize the control of UDS [4]. In [12], evolutionary algorithms (EA) together with self-adaptive strategies get an increase in the quality of the water reaching the rivers, reducing costs. Control algorithms based on population dynamics (PD) have been used in [18,19], getting a better use of the sewage capacity and reducing overflows. All the above control strategies are based on sewer network hydraulic models, except in [20,21], in which the concentration of wastewater pollutants is considered in the control algorithm. A literature review of modelling and control of sewer systems is presented in [22].

Related to the structure of the UDS control systems, centralized control is the most used configuration. However, decentralized control using local controllers may be a better solution if the number of actuators is high [23]. In complex large-scale systems, both levels of control (local and global) are usually considered together and frequently, and there can be one more level of control, becoming a hierarchical structure [24,25].

The interaction between the UDS and the WWTP must be considered to improve performance in both systems [26], but there are only few studies in the literature that tackle both systems together and, mainly, they deal with the performance of the WWTP, using simpler models for the sewage systems [27,28]. Considering other control algorithms, in [29] is used a heuristic-type controller based on rules, and in [30], another that combines model predictive control (MPC) with RBC strategies.

One of the most popular advanced control strategies is MPC that uses a system model to obtain the control variables values on a future horizon by optimizing a cost function [31–33]. The MPC algorithm consists of four main parts: a control-oriented prediction model of the process, a cost function representing the control objective, a group of process constraints, and a minimization problem solved in a receding-horizon way [31]. Generally, hierarchical control architectures use the MPC to calculate optimal set-points for local controllers. The features of MPC strategies have several advantages for being used in UDS, such as their ability to predict the system's behavior to future rain events considering delays, constraints, and disturbances. A centralized predictive controller, applied in [25], has been developed considering a model whose state variables are the levels of the network reservoirs, the manipulated variables are flow rates at the tank outputs, and the measurable disturbances are the collected input flow rates to the system. In this case, the main constraints are given by the maximum capacity of tanks and pipes.

In [25,34], an MPC controller has been simulated, obtaining important decreases in floods and overflows. In other cases, MPC algorithms have been applied to UDS by means of non-linear prediction models and both operation costs and overflows have decreased [35,36]. Most parts of the control-oriented models in MPCs consider the hydraulic part only. However, in [20,37], a dynamic model of concentration of suspended solids (TSS) used for the MPC algorithm to minimize not only the CSO volume, but also the total pollutant mass escaping into the environment, is presented. Moreover, in [38], a comparison of two optimization methods to minimize overflows is shown: mixed integer (MI) and quadratic program (QP). This analysis concludes that both methods have the same performance, but QP is computationally more efficient than MI.

In the case of large-scale systems, it can be more suitable to split the whole process into smaller subsystems to simplify the application of MPC controller [9,39]. Then, local prediction models and cost functions are used, and the exchange of information between subsystems may be considered or not. Distributed model predictive control (DMPC) [9]

is applied when local controllers exchange data to solve their own local problem, in a cooperative and coordinated way. In the literature, there is a wide variety of DMPC algorithms applied to different processes [40,41], but only a few deal with water management systems, such as volume control in tanks [42,43] and the coordination of drinking water systems [41,44]. In addition, no applications to sewer systems based on DMPC algorithms considering the concentrations of pollutants have been developed before, to the knowledge of the authors.

Therefore, the major contribution of this article is the development and application of a practical DMPC algorithm to a UDS, using local linearized models of the process including TSS and fuzzy negotiation among the subsystems [42,43]. Moreover, the WWTP is included in the control algorithm as an additional objective which consists of the maximization of the WWTP input flow. The results, obtained from the benchmark described in [5], have been compared with a centralized MPC and DMPC based in a cooperative game [45] to validate the utility of the proposed methodology, providing a comprehensive framework for validation including pollutants and other practical issues.

This document begins with an introduction describing the benchmark simulation model and presenting the model used by the control algorithms. In the next section, its sectorization is detailed. This article follows with the exposition of the control objectives. Then, centralized and distributed MPC control strategies are explained, showing the compared results in each case considering or not considering TSS. Finally, the conclusions of this work are presented.

## 2. Benchmark Model and Evaluation Criteria

### 2.1. Benchmark Model Description

The model describes, in a realistic way, the operation of a sewer network, being excellent for testing different control strategies that optimize the performance of the system. This benchmark simulation model for a sewer system [5] has algorithms to produce different scenarios that include both the features and the volume of urban wastewater and the runoff reaching the sewer network, generated by rainfall of variable duration and intensity, considering the season of the year or day of the week. The generation of wastewater at each catchment area combines several contributions: domestic, industry, stormwater, and infiltration to sewage. The pollutants included are chemical oxygen demand (COD), which can be divided into soluble COD and particulate COD; ammonia ($NH_4^+$); nitrate ($NO_3^-$); and phosphate ($PO_4^{3-}$). Particulate COD is the main component of suspended solids. There are six catchment areas (Figure 1), numbered from 1 to 6.

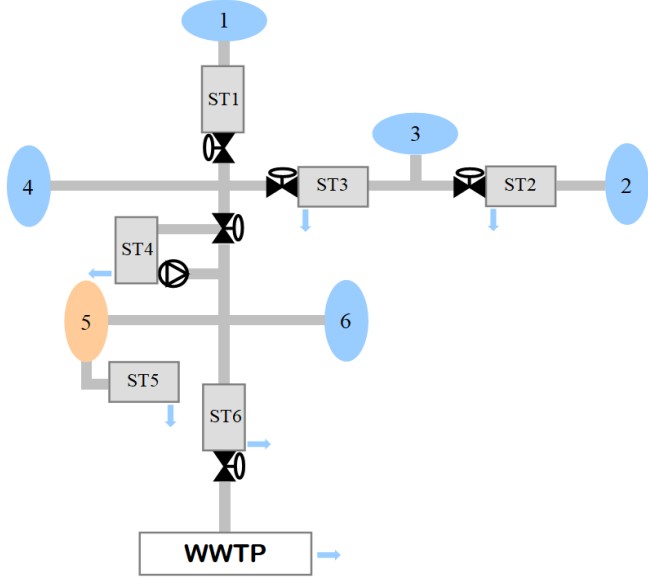

**Figure 1.** Catchment and sewer network model.

The sewer model consists of three elements: a transport submodel describing the evolution of both flow rate and pollutants in the sewer, a first flush submodel representing the sudden increase of particulates when the rain starts after a long period of drought, and several types of storage tank submodels. This model, shown in Figure 1, is made up of six storage tanks, named ST1, ..., ST6 (ST5 is an off-line tank); water links; a pump and five valves for flow control; and a wastewater treatment plant.

The major objective of the sewage network is to catch all the urban residual water and lead it to the WWTP, keeping constant a supply flow as close as possible to its nominal inflow rate. This can be obtained by holding the wastewater in the tanks and releasing when the inflow of the WWTP decreases. Moreover, another objective is to minimize the overflows in the deposits and at the inlet of WWTP when rainfall is so intense that it can cause flooding and discharges of contaminant substances into the environment.

As a part of the process of design and assessment of the controllers, a simplified model of the system has been developed and used for the control algorithm. The simplifications consist of considering the hydraulic part of the benchmark together with the concentration of suspended solids TSS, disregarding variables that represent concentrations of other pollutants. The ST5 tank has been eliminated because in this configuration, it is not controllable. The ST6 deposit has been renumbered as 5. In addition, the ST4 reservoir is considered like the rest because the inlet valve is fixed, and the outlet flow is generated by opening a valve instead of activating the pump. In consequence, the mathematical model of the system contains the following parts [5]:

*Water catchment areas*: zones when the water is collected constituting the inlet flow to the sewage that is considered as a disturbance. The resultant flow in each zone are represented by $q_{ri}$, $i = 1, \ldots, 6$.

*Link elements*: gravity wastewater pipes in an open channel. They connect the resultant flows in each zone with the deposits, and the deposits with each other and with the WWTP. They are represented as first-order systems with very slow dynamics, depending on their length, including the concentration of total suspended solids, as shown in [37]. Their discrete mathematical model is:

$$q_i(k+1) = \left(1 - \frac{T}{\tau_i}\right)q_i(k) + \left(\frac{T}{\tau_i}\right)q_{u,i}(k),$$

$$Tss_{out,i}(k+1) = (1 - c_i)Tss_{out,i}(k) + c_i Tss_{in,i}(k), i = 1, 2, 3 \ldots, 9,$$

(1)

where:

$T$ is the sampling period;
$\tau_i$ is the time constant of the element $i$;
$q_i(k)$ is the output flow of the element $i$;
$Tss_{out,i}$ is the concentration of suspended solids in the output flow of the element $i$;
$q_{u,i}(k)$ is the sum of inflows to the link element $i$;
$Tss_{in,i}$ is the concentration of suspended solids in the input flow of the element $i$;
$c_i$ is the sedimentation coefficient of suspended in the element $i$ (parameter that needs calibration);

*Storage tanks*: deposits for storing wastewater. Wastewater stored can overflow if a maximum tank volume is overcome. Their discrete model for the water level and TSS is:

$$h_i(k+1) = h_i(k) + \frac{T}{A_i}[u_{in,i}(k) - u_i(k) - q_{ov,i}(k)]$$

$$u_i(k) = a_i(k)c_{0i}\sqrt{h_i(k)}, \; V_i(k) = A_i h_i(k), \; V_{\max,i} = A_i h_{\max,i}$$

(2)

$$Tss_{out,i}(k+1) = (1 - c_i)Tss_{out,i}(k) + c_i Tss_{in,i}(k), i = 1, 2, 3, 4, 5,$$

where all parameters are related to tank $i$ and instant $k$:

$V_{\max,i}$ is the maximum capacity of the tank;
$V_i(k)$ is the filled volume;
$Tss_{in,i}$ is the concentration of suspended solids in the input flow of the tank $i$;
$Tss_{out,i}$ is the concentration of suspended solids in the output and overflow flow of the tank $i$;

$u_{in,i}(k)$ is the input flow rate;

$u_i(k)$ is the output flow rate;

$q_{ov,i}(k)$ is the overflow flow rate;

$c_{0i}$ is the discharge coefficient (experimental parameter depending on the tank $i$);

$c_i$ is the sedimentation coefficient of suspended in the tank $i$ (parameter that needs calibration);

$A_i$ is the tank area;

$h_{max,i}$ is the tank height;

$h_i(k)$ is the water level;

$a_i(k)$ is the opening of the deposit outlet valve (control variable: $a_i \in [0, 1]$);

*Nodes*: they are places of union of several water pipes. The output flow is the addition of the input flows, and the concentration of suspended solids at the outlet is obtained assuming that the mass of suspended solids is conserved in the node [37]. For example, for a node where three pipes join:

$$q_{in3}(k) = q_{out1}(k) + q_{out2}(k)$$

$$q_{in3}(k)Tss_{in3}(k) = q_{out1}(k)Tss_{out1}(k) + q_{out2}(k)Tss_{out2}(k) \tag{3}$$

$$Tss_{in3}(k) = \frac{q_{out1}(k)Tss_{out1}(k) + q_{out2}(k)Tss_{out2}(k)}{q_{out1}(k) + q_{out2}(k)},$$

In Figure 2, the node model of the example is shown.

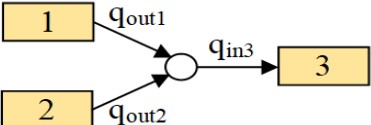

**Figure 2.** Node model.

Therefore, the expression that allows the calculation of the concentration of solids at the outlet of a node has a non-linear relationship with the flows and concentrations of solids at the inlet of the node, which will also be considered as state variables of the process. As such, to get a linear model of this system, which can be part of the control algorithm, we have to linearize the following expression:

$$Tss_{in3} = \frac{q_{out1}Tss_{out1} + q_{out2}Tss_{out2}}{q_{out1} + q_{out2}} \approx K_{qout1}q_{out1} + K_{qout2}q_{out2} + K_{Tssout1}Tss_{out1} + K_{Tssout2}Tss_{out2},$$

$$\text{where}: K_{qout1} = \frac{q_{out2o}(Tss_{out1o} - Tss_{out2o})}{(q_{out1o} + q_{out2o})^2}, K_{qout2} = \frac{q_{out1o}(Tss_{out2o} - Tss_{out1o})}{(q_{out1o} + q_{out2o})^2} \tag{4}$$

$$K_{Tssout1} = \frac{q_{out1o}}{q_{out1o} + q_{out2o}}, K_{Tssout2} = \frac{q_{out2o}}{q_{out1o} + q_{out2o}},$$

are coefficients calculated for each sampling time at the point of operation from the values of the flow rates and concentrations of solids measured at that point.

The whole system is shown in Figure 3, depicted by a block diagram made of these simple elements: catchments, represented by ovals; storage tanks, by triangles; and link elements, by rectangles. This diagram is like the benchmark represented in Figure 1, but more detailed.

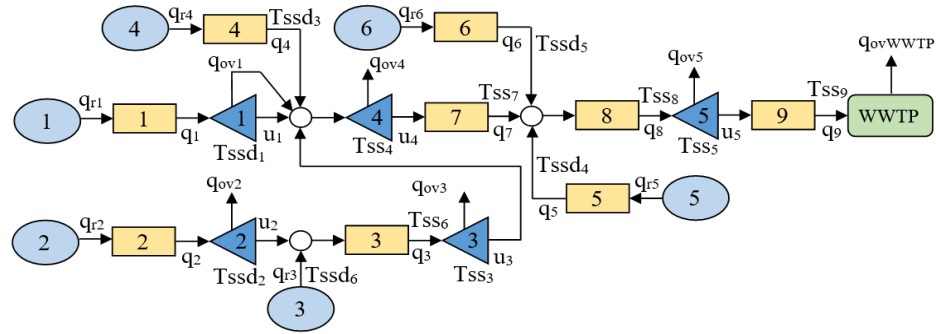

**Figure 3.** Simplified sewer block diagram.

The state variables are the tank levels 1, 2, 3, 4, and 5 ($x_1, \ldots, x_5$); the flow rates of the pipes 3, 7, 8, and 9, corresponding to the states ($x_6, \ldots, x_9$); and the concentrations of suspended solids in each of them, respectively, except in tanks 1 and 2, since in these cases, the concentrations are not controllable ($x_{10}, \ldots, x_{16}$) because the inlet flow to these tanks cannot be controlled and this is the only way to modify their concentrations. Moreover, the output flow of pipe 9 (state $x_9$) is the inlet flow to the WWTP if it does not exceed its nominal value. The output flows of the pipes of the catchments 1, 2, 4, 5, and 6, as well as the flow caught in zone 3, will represent measurable disturbances on the system ($d_1, \ldots, d_6$), as well as the concentrations of suspended solids in tanks 1 and 2, in the outlet flows of the link elements of the catchment areas 4, 5, and 6, and in the flow caught in zone 3 ($d_7, \ldots, d_{12}$). The overflowed volume in reservoir 1 returns to the sewer through tank 4. The control variables are the desired output flows of the tanks. They also are the set-points of local PID flow controllers at a lower level ($u_1, \ldots, u_5$). Therefore, the state vector, the control variables vector, and the disturbances vector are defined considering the benchmark variables as:

$$\mathbf{x} = (h_1, h_2, h_3, h_4, h_5, q_3, q_7, q_8, q_9, Tss_3, \ldots, Tss_9), \quad \mathbf{u} = (u_1, u_2, u_3, u_4, u_5),$$

$$\mathbf{d} = (q_1, q_2, q_4, q_5, q_6, q_{r3}, Tssd_1, \ldots, Tssd_6),$$

(5)

Table 1 shows the system variables and their correspondence with the considered variables of the state space model.

**Table 1.** System variables.

| Space State Model Variables | System Variables | Meaning and Units |
|:---:|:---:|:---:|
| $x_1, \ldots, x_5$ | $h_1, \ldots, h_5$ | Tank levels (m) |
| $x_6, x_7, x_8, x_9$ | $q_3, q_7, q_8, q_9$ | Pipes 3, 7, 8, and 9 flow rates (m$^3$/d) |
| $x_{10}, x_{11}, x_{12}$ | $Tss_3, Tss_4, Tss_5$ | Suspended solids in tanks 3, 4, and 5 (g/m$^3$) |
| $x_{13}, x_{14}, x_{15}, x_{16}$ | $Tss_6, Tss_7, Tss_8, Tss_9$ | Suspended solids in pipes 3, 7, 8, and 9 (g/m$^3$) |
| $u_1, \ldots, u_5$ | $u_1, \ldots, u_5$ | Tank output flow rates (m$^3$/d) |
| $d_1, d_2, d_3, d_4, d_5, d_6$ | $q_1, q_2, q_4, q_5, q_6, q_{r3}$ | Catchment flow rates (m$^3$/d) |
| $d_7, \ldots, d_{12}$ | $Tssd_1, \ldots, Tssd_6$ | Catchment suspended solids (g/m$^3$) |

The mathematical model, in terms of discrete differential equations, is:

$$x_1(k+1) = x_1(k) + \tfrac{T}{A_1}[d_1(k) - u_1(k) - q_{ov1}(k)]$$

$$x_2(k+1) = x_2(k) + \tfrac{T}{A_2}[d_2(k) - u_2(k) - q_{ov2}(k)]$$

$$x_3(k+1) = x_3(k) + \tfrac{T}{A_3}[x_6(k) - u_3(k) - q_{ov3}(k)]$$

$$x_4(k+1) = x_4(k) + \tfrac{T}{A_4}[d_3(k) + u_1(k) + q_{ov1}(k) + u_3(k) - u_4(k) - q_{ov4}(k)]$$

$$x_5(k+1) = x_5(k) + \tfrac{T}{A_5}[x_8(k) - u_5(k) - q_{ov5}(k)]$$

$$x_6(k+1) = \left(1 - \tfrac{T}{\tau_3}\right)x_6(k) + \left(\tfrac{T}{\tau_3}\right)[d_6(k) + u_2(k)]$$

$$x_7(k+1) = \left(1 - \tfrac{T}{\tau_7}\right)x_7(k) + \left(\tfrac{T}{\tau_7}\right)u_4(k)$$

$$x_8(k+1) = \left(1 - \tfrac{T}{\tau_8}\right)x_8(k) + \left(\tfrac{T}{\tau_8}\right)[d_4(k) + d_5(k) + x_7(k)]$$

(6)

$$x_9(k+1) = \left(1 - \tfrac{T}{\tau_9}\right)x_9(k) + \left(\tfrac{T}{\tau_9}\right)u_5(k)$$

$$x_{10}(k+1) = (1 - c_3)x_{10}(k) + c_3 x_{13}(k)$$

$$x_{11}(k+1) = (1 - c_4)x_{11}(k) + c_4 \tfrac{(u_1(k) + q_{ov1}(k))d_7(k) + d_3(k)d_9(k) + u_3(k)x_{10}(k)}{u_1(k) + q_{ov1}(k) + d_3(k) + u_3(k)}$$

$$x_{12}(k+1) = (1 - c_5)x_{12}(k) + c_5 x_{15}(k)$$

$$x_{13}(k+1) = (1 - c_6)x_{13}(k) + c_6 \tfrac{u_2(k)d_8(k) + d_6(k)d_{12}(k)}{u_2(k) + d_6(k)}$$

$$x_{14}(k+1) = (1 - c_7)x_{14}(k) + c_7 u_4(k)$$

$$x_{15}(k+1) = (1 - c_8)x_{15}(k) + c_8 \tfrac{d_5(k)d_{11}(k) + x_7(k)x_{14}(k) + d_4(k)d_{10}(k)}{d_5(k) + x_7(k) + d_4(k)}$$

$$x_{16}(k+1) = (1 - c_9)x_{16}(k) + c_9 u_5(k),$$

Then, a linear model is obtained from this model, removing the overflow flow rate terms and linearizing the nonlinear equations in the nodes. This will be used as a prediction model in the MPC strategy:

$$x_{11}(k+1) = (1-c_4)x_{11}(k) + c_4[K_{4Tss1out}d_7(k) + K_{4Tss3out}x_{10}(k) +$$
$$+ K_{4u1}u_1(k) + K_{4u3}u_3(k) + K_{4q4}d_3(k) + K_{4Tssd3}d_9(k)]$$
$$x_{13}(k+1) = (1-c_6)x_{13}(k) + c_6[K_{6Tss2out}d_8(k) + K_{6u2}u_2(k) +$$
$$+ K_{6qr3}d_6(k) + K_{6Tssd6}d_{12}(k)]$$
$$x_{15}(k+1) = (1-c_8)x_{15}(k) + c_8[K_{8q7}x_7(k) + K_{8Tss7out}x_{14}(k) +$$
$$+ K_{8q5}d_4(k) + K_{8q6}d_5(k) + K_{8Tssd4}d_{10}(k) + K_{8Tssd5}d_{11}(k)]$$

(7)

Hence, the state space model equations are:

$$\mathbf{x}(k+1) = \mathbf{A}\mathbf{x}(k) + \mathbf{B}_p\mathbf{u}(k) + \mathbf{B}_d\mathbf{d}(k) \Rightarrow \mathbf{x}(k+1) = \mathbf{A}\mathbf{x}(k) + \mathbf{B}\begin{bmatrix} \mathbf{u}(k) \\ \mathbf{d}(k) \end{bmatrix},$$

$$\text{where}: \ \mathbf{B} = \begin{bmatrix} \mathbf{B}_p & \mathbf{B}_d \end{bmatrix}, \ \mathbf{x} = (x_1, x_2, \dots, x_{16}), \mathbf{d} = (d_1, d_2, \dots, d_{12}),$$

$$\mathbf{u} = (u_1, u_2, \dots, u_5),$$

(8)

The matrices of the model are shown at the end of this paper in Appendix A.

### 2.2. Evaluation Criteria

The evaluation of the behavior of the system includes different places of overflow in the sewage and at the input of the WWTP. The evaluation criteria used to analyze the effects of the applied control strategies are shown below. Only hydraulic variables are considered in some of them, whereas others also include magnitudes related to water quality. Some of the indices, such as $N_{ov}$ (number of overflows), $T_{ov}$ (duration of overflow), $V_{ov}$ (overflow volume), $G_u$ (degree of usage of WWTP), and $S$ (smoothness in control law application), corresponding to the first group, are shown in [46]. In this work, the following qualitative indices will be considered, where the subscript $i$ stands for a specific overflow point in the system and $n$ is the total number of points:

1.  Total suspended solids mass ($Mss_{ov,i}$) (kg): this is the total mass of suspended solids at a specific overflow place $i$. Considering simulation time in days (d) $T_{sim}$, if $q_{ov,i}(t)$ is the overflow (m$^3$/d) and $Tss_i(t)$ is the total suspended solids concentration (g/m$^3$), the total suspended solids mass of pollutant overflowed at the point $i$ is:

$$Mss_{ov,i} = 10^{-3} \int_0^{T_{sim}} q_{ov,i}(t)Tss_i(t)dt,$$

(9)

and the total mass of suspended solids overflowed in the whole sewer is:

$$Mss_{ov} = \sum_{i=1}^{n} Mss_{ov,i},$$

(10)

2.  Ammonia mass ($NH_{ov,i}$) (kg): this is the total mass of ammonia in wastewater escaping from the sewage at a specific overflow place $i$. If $NH_i(t)$ is the ammonia concentration, the total ammonia mass overflowed at the point $i$ is:

$$NH_{ov,i} = 10^{-3} \int_0^{T_{sim}} q_{ov,i}(t)NH_i(t)dt,$$

(11)

and the total mass of ammonia overflowed in the whole sewer is:

$$NH_{ov} = \sum_{i=1}^{n} NH_{ov,i}, \tag{12}$$

3.　Nitrate mass ($NO_{ov,i}$) (kg): this is the total mass of nitrate in wastewater discharged from a specific overflow place $i$. If $NO_i(t)$ is the nitrate concentration, the total nitrate mass overflowed at the point $i$ is:

$$NO_{ov,i} = 10^{-3} \int_{0}^{T_{sim}} q_{ov,i}(t) NO_i(t) dt, \tag{13}$$

and the total mass of nitrate overflowed in the whole sewer is:

$$NO_{ov} = \sum_{i=1}^{n} NO_{ov,i}, \tag{14}$$

4.　Phosphate mass ($PO_{ov,i}$) (kg): this is the total mass of phosphate in wastewater escaping from the sewer network at a certain overflow place $i$. If $PO_i(t)$ is the phosphate concentration, the total phosphate overflowed at the point $i$ is:

$$PO_{ov,i} = 10^{-3} \int_{0}^{T_{sim}} q_{ov,i}(t) PO_i(t) dt, \tag{15}$$

and the total mass of phosphate overflowed in the whole sewer is:

$$PO_{ov} = \sum_{i=1}^{n} PO_{ov,i}, \tag{16}$$

5.　Overflow quality index ($OQI_i$) (kg-pollution units/d): this is an aggregated index representing the total mass of pollutants in wastewater discharged into treated water receivers from a determined overflow place $i$ during a simulation time $T_{sim}$. It includes all the pollutants with different weights, like those used in BSM2 for WWTP [22]. In this case, $OQI_i$ can be approximated as:

$$OQI_i = \frac{1}{T_{sim}} [w_{Tss} Mss_{ov,i} + w_{NH} NH_{ov,i} + w_{NO} NO_{ov,i} + w_{PO} PO_{ov,i}],$$
$$\text{and the total index}: \ OQI = \sum_{i=1}^{n} OQI_i, \tag{17}$$

with $w_{Tss} = 2$, $w_{NH} = 30$, $w_{NO} = 10$, and $w_{PO} = 100$.

### 3. Sectorization of the System Model

To apply control methodologies to large-scale systems, it is usually necessary to divide the whole system into smaller subsystems. To find the best way to divide the system into smaller ones, a structural analysis has been carried out, so that the subsystems are controllable, reducing their degree of coupling [47]. The direct graph of the system represents the relationships between the different process variables and helps to obtain the best method to split the entire system into subsystems with minimal coupling, holding their reachability [47]. To achieve this objective, the reachability from the input has been tested from the direct graph of the system obtained by applying the same method as in [46]. This is shown in Appendix B of the document, as well as the system sectorization.

For this benchmark, two subsystems are considered: the first includes tanks 1 to 4, links 1 to 4, and catchment areas 1 to 4; and the second, tank 5, links 5 to 9, and catchment areas 5 and 6. To use the distributed MPC exposed in Section 6, the state space local models of each subsystem are obtained as follows, considering the coupling input $u_4$ to belong to subsystem 1:

$$\mathbf{x}_1(k+1) = \mathbf{A}_1\mathbf{x}_1(k) + \mathbf{B}_{p11}\mathbf{u}_1(k) + \mathbf{B}_{p12}\mathbf{u}_2(k) + \mathbf{B}_{d11}\mathbf{d}_1(k) + \mathbf{B}_{d12}\mathbf{d}_2(k),$$

$$\mathbf{y}_1(k) = \mathbf{C}_1\mathbf{x}_1(k), \text{ where : } \mathbf{x}_1 = (x_1, x_2, x_3, x_4, x_6, x_{10}, x_{11}, x_{13}), \mathbf{u}_1 = (u_1, u_2, u_3, u_4),$$

$$\mathbf{d}_1 = (d_1, d_2, d_3, d_6, d_7, d_8, d_9, d_{12}),$$

$$\mathbf{x}_2(k+1) = \mathbf{A}_2\mathbf{x}_2(k) + \mathbf{B}_{p21}\mathbf{u}_1(k) + \mathbf{B}_{p22}\mathbf{u}_2(k) + \mathbf{B}_{d21}\mathbf{d}_1(k) + \mathbf{B}_{d22}\mathbf{d}_2(k),$$

$$\mathbf{y}_2(k) = \mathbf{C}_2\mathbf{x}_2(k), \text{ where : } \mathbf{x}_2 = (x_5, x_7, x_8, x_9, x_{12}, x_{14}, x_{15}, x_{16}),$$

$$\mathbf{u}_2 = (u_5), \ \mathbf{d}_2 = (d_4, d_5, d_{10}, d_{11}),$$

$$(18)$$

Properly choosing the rows and columns of **A** and **B** from the matrices (8), the matrices $\mathbf{A}_1$, $\mathbf{B}_{p11}$, $\mathbf{B}_{p12}$, $\mathbf{A}_2$, $\mathbf{B}_{p21}$, and $\mathbf{B}_{p22}$ are obtained. Moreover, due to the system's configuration, $\mathbf{B}_{p21}$ only has non-zero elements in the last column, and $\mathbf{B}_{p12}$, $\mathbf{B}_{d12}$, and $\mathbf{B}_{d21}$ are null.

## 4. Control Objectives

The control objectives considered in this system are to keep the WWTP input flow close to its nominal value, taking advantage of its capacity, to reduce the mass of contaminant escaping from the sewage, avoiding overflows in the deposits and at the input of the WWTP, and to optimize operating costs. To get the exposed goal, the set-points of the outlet flow rates of the deposits (control variables) are obtained to reduce the difference between the nominal flow rate and its current value at the inlet to the WWTP. Moreover, the volume of wastewater is distributed among all the reservoirs in the sewerage according to their capacity, which is obtained by optimizing the difference between the wastewater level of each deposit and a dynamically calculated set-point level to get that objective [48]. The effects of the disturbances (collected flows at the catchment areas) will be reduced by the proportional distribution of the wastewater stored in the reservoirs, minimizing the overflows. The control objectives can be expressed as a cost function that includes the partial goals previously exposed [46,49]:

$$J = \sum_{j=1}^{M} \sum_{k=0}^{N} w_j \varphi_j(\mathbf{x}(k), \mathbf{u}(k)) \tag{19}$$

where $N$ is the prediction horizon of the MPC presented in 5 and $M$ is the number of objectives considered, $\varphi_j$ is a partial goal, and $w_j$ is the weight associated to each partial objective $\varphi_j$, with $j = 1, \dots, M$.

The partial objectives included in the control problem considered are shown below:

1. Minimization of load overflowed and overflows, and uniform distribution of the stored wastewater and the concentration of total suspended solids:

$$\varphi_1(\mathbf{x}(k)) = \sum_{i=1}^{5} q_{ii}(k)(V_i(k) - v_i V_G(k))^2 + \sum_{j=10}^{12} q_{jj}(k)\left(Tss_{j-7}(k) - Tss_m(k)\right)^2,$$

$$v_i = \frac{V_{i\max}}{\sum\limits_{j=1}^{5} V_{j\max}}, \ V_G(k) = \sum_{j=1}^{5} V_j(k),$$

$$\text{with} q_{ii}(k) = \begin{cases} f_i & x_i(k) \le h_{\max i} \\ f_i\left(1 + \alpha_i(x_i(k) - h_{\max i})^2 \frac{Tss_i(k)}{Tss_m(k)}\right) & x_i(k) > h_{\max i} \end{cases}, \tag{20}$$

$$Tss_m(k) = \frac{\sum\limits_{j=1}^{5} V_j(k) Tss_j(k)}{\sum\limits_{j=1}^{5} V_j(k)}$$

where $V_G$ is the total filled volume in the sewerage at instant $k$, $v_i$ is a factor that represents the weight of the deposit capacity $i$ in the total available storage volume in the network,

$Tss_i(k)$ represents the concentration of suspended solids of each tank $i$ at the instant $k$, and $Tss_m(k)$ is the weighted mean concentration of suspended solids at the instant $k$. The weight $q_{ii}(k)$ allows for the overflows' penalization, growing quadratically with the associated overflow, and also considers if the pollutant concentration $Tss_i(k)$ is higher or lower than the average $Tss_m(k)$ to penalize more or less the mass of pollutant overflowed. Factors $f_i$ and $\alpha_i$ penalize even more overflows in some deposits. Weights $q_{ii}(k)$ and $q_{jj}(k)$ are tuning parameters included as diagonal elements in the $\mathbf{Q}(k)$ matrix below.

2. Maximum usage and minimum overflow at the WWTP influent:

$$\varphi_2(\mathbf{x}(k)) = (Q_{WWTP}(k) - Q_{WWTP\text{max}})^2 \tag{21}$$

where $Q_{WWTP}$ is the inlet flow to the treatment plant (state $x_9$) at instant $k$ and $Q_{WWTP\text{max}}$ is its nominal value.

3. Control efforts minimization:

$$\varphi_3(\mathbf{u}(k)) = \sum_{i=1}^{5} r_{ii}\left(u_i(k) - u_{iref}(k)\right)^2 \tag{22}$$

where $u_{iref}$ are the output flows to keep the reference level in a deposit, calculated by Bernoulli's law, as explained in Section 5.2. Weights $r_{ii}$ are tuning parameters included as diagonal elements in the $\mathbf{R}$ matrix below.

The global control system has a hierarchical structure. The level set-points are generated for each variable at the upper level to get the cited control objectives. Thus, the MPC controller solves a constrained optimization problem, obtaining the set-points used by local controllers, optimizing the usage of WWTP and operating costs. Then, local controllers apply the control signals to the system [48,50].

## 5. Predictive Control Problem with Online Linearization

### 5.1. Optimization Problem

In this work, the centralized predictive control algorithm considers a linear state-space prediction model, including the influence of disturbances. This model changes every time new values of the process variables are taken, due to the linearization of the expressions in each node, so matrices A and B depend on the sample instant $k$. The key point is that the optimization of (23) is equivalent to minimize Equation (16), where different objectives are considered, by using specific weights $\mathbf{Q}(k)$ and $\mathbf{R}$ in the MPC problem as explained below, related to weight $w_j$ in (19) and the specific performance indices (20), (21), and (22).

The objective function of the MPC contains both the control errors in the system states and the difference between the control sequence and the flow set-point (hence, it penalizes control energy), considering the control and prediction horizons are equal to $N$:

$$J(k, \mathbf{U}) = \sum_{i=1}^{N-1} \left[\|\hat{\mathbf{x}}(k+i) - \mathbf{x}_{ref}(k)\|_{\mathbf{Q}(k)}^2 + \|\mathbf{u}(k+i) - \mathbf{u}_{ref}(k)\|_{\mathbf{R}}^2\right] + \mathbf{x}^T(k+N)\mathbf{P}\mathbf{x}(k+N), \tag{23}$$

where $\mathbf{x}_{ref}(k) = (x_{1ref}, x_{2ref}, \ldots, x_{16ref})$ and $\mathbf{u}_{ref}(k) = (u_{1ref}, u_{2ref}, \ldots, u_{5ref})$ are the states and inputs set-points, and $\mathbf{U}(k) = \begin{bmatrix} \mathbf{u}(k) & \mathbf{u}(k+1) & \ldots & \mathbf{u}(k+N-1) \end{bmatrix}^T$.

For the MPC algorithm, the optimization problem is:

$$\mathbf{U}^*(k) = \underset{\mathbf{U}(k)}{\arg\min} J(k), \ \mathbf{U}^*(k) = \begin{bmatrix} \mathbf{u}^*(k) & \mathbf{u}^*(k+1) & \ldots & \mathbf{u}^*(k+N-1) \end{bmatrix}^T \text{subject to}:$$

$$\hat{\mathbf{x}}(k+i+1) = \mathbf{A}(k)\hat{\mathbf{x}}(k+i) + \mathbf{B}_p(k)\mathbf{u}(k+i) + \mathbf{B}_d(k)\mathbf{d}(k+i), \ \mathbf{d}(k+i) = \mathbf{d}(k)$$

$$\hat{\mathbf{x}}(k) = \mathbf{x}(k) \tag{24}$$

$$0 \leq \hat{x}_j(k+i), \ i = 0, \ldots, N-1, \ j = 1, \ldots, 5$$

$$0 \leq \hat{x}_j(k+i) \leq q_{\text{max}j}, \ i = 0, \ldots, N-1, \ j = 6, \ldots, 9$$

$$0 \leq u_j(k+i) \leq u_{\text{max}j}, \ i = 0, \ldots, N-1, \ j = 1, \ldots, 5,$$

where $q_{\mathrm{max}j}$ and $u_{\mathrm{max}j}$ are the upper bounds for flow rate in the link elements and the tank outputs, respectively.

For the MPC algorithm, the horizon $N$, matrices $\mathbf{Q}(k)$ and $\mathbf{R}$, and a terminal penalty for MPC stability, matrix $\mathbf{P}$, obtained from the Riccati equation [51], are tuning parameters. A variable diagonal matrix $\mathbf{Q}(k)$ includes all control objectives of 4. When an overflow occurs in a deposit, the associated weight grows to further penalize the difference with the level set-point of this deposit. The elements $q_{11}, \ldots, q_{55}, q_{99}, \ldots, q_{12,12}$ are the non-zero elements of $\mathbf{Q}(k)$ and are calculated by (20), where $f_i$ and $\alpha_i$ are parameters for tuning MPC. All the rest of the elements are zero because they are related to state variables of pipes and their suspended solids concentrations that do not need to be optimized. To penalize the deviations of the flow set-points in respect to their reference and minimize the operating costs, a diagonal matrix $\mathbf{R}$ is used.

This problem is a quadratic optimization problem (*QP*) with constraints [31]. The developed algorithm holds the last calculated solution, if the optimization problem is not feasible, to avoid failure of the control system.

*5.2. Set-Point Determination*

Optimal operation can be achieved by a hierarchical controller that calculates the reservoirs' level set-points to distributing the current volume of wastewater among all the deposits considering their maximum capacity [48]. Therefore, it is necessary to calculate for each one its reference level, considering the total capacity of the system and the volume of that tank at each sampling instant:

$$x_{iref}(k) = \left( \frac{V_G(k)}{A_i} \right) v_i, i = 1, \ldots, 5, \tag{25}$$

where $x_{iref}(k)$ is the set-point level for deposit $i$ at instant $k$. Moreover, the nominal input flow to the WWTP is $x_{9ref} = 60,000$ m$^3$/d. The reference values for the states $x_{ref10}, \ldots, x_{ref12}$, representing pollutant concentrations, are all taken equal to the average value of the pollutant concentration at the instant considered $Tss_m(k)$. The references for the rest of states are irrelevant because the corresponding weight in $\mathbf{Q}$ matrix is 0.

The flow rate set-points are obtained considering the set-point level for each deposit, according to expression:

$$u_{iref}(k) = c_{0i}\sqrt{x_{iref}(k)}, \ i = 1, \ldots, 5, \tag{26}$$

## 6. DMPC and Fuzzy Negotiation

*6.1. Distributed Model Predictive Control (DMPC) with Fuzzy Negotiation*

The DMPC strategy presented in this article is based on [52]. It consists of the optimization of local problems including the future behaviour of the inputs of each subsystem and the neighbour, using local constraints and prediction models. The local MPC problems are analogous to the one presented in Section 5.1, adding neighbouring constraints. Two agents compute each optimization problem. Each agent obtains a control sequence for their subsystem, $\mathbf{U}_1^*$ and $\mathbf{U}_2^*$, at every sampling period, keeping the neighbour control sequence constant to $\mathbf{U}_1^s$ and $\mathbf{U}_2^s$, respectively, where $\mathbf{U}_1^s$ and $\mathbf{U}_2^s$ are the previous solutions of optimization problem extended to the current $k$. Then, another optimization problem is solved by agent 2, providing the control law for the neighbour $\mathbf{U}_1^w$, keeping their constant, as shown in [46]. Note that agent 1 does not solve any additional optimization problem to provide $\mathbf{U}_2^w$ because $\mathbf{U}_2$ does not affect subsystem 1 because input $u_5$ belongs to subsystem 2 only. The fuzzy negotiation provides the final control sequences for subsystem 1, $\mathbf{U}_1^f$ considering the two possible solutions available $\mathbf{U}_1^*$ and $\mathbf{U}_1^w$, because $\mathbf{U}_2^f = \mathbf{U}_2^*$. To apply the fuzzy negotiation, the average input flow to the WWTP, $\bar{q}_{WWTP}$, and the TSS mass overflowed on average for all tanks in the prediction horizon $\overline{m}_{ssov}$ are obtained for each of the calculated control sequences. Local prediction models, local solutions, and disturbances are considered for this:

$$\overline{q}_{WWTP}(k) = \frac{1}{N}\sum_{i=1}^{N}\hat{x}_9(k+i), \overline{m}_{ssov}(k) = \frac{1}{N}\sum_{j=2}^{5}\sum_{i=1}^{N}\hat{q}_{ov,j}(k+i) \cdot \hat{T}ss_j(k+i), \quad (27)$$

### 6.2. Fuzzy Negotiation

The fuzzification process consists of obtaining fuzzy sets from imprecise knowledge of a system or process. Then, fuzzy values are obtained considering numerical values of a certain property. Defuzzification allows the calculation of numerical values of certain output variables applying a rule set to the inputs and interpreting the membership degrees of the fuzzy sets in a specific decision [53,54]. This method is used in a DMPC algorithm with negotiation between agents to get the best solution to the whole system [46].

Fuzzy sets for $\overline{q}_{WWTP}$ and $\overline{m}_{ssov}$ have been considered with trapezoidal shape and obtained in a heuristic way in order to get the best performance of the entire system (Figure 4). In particular, the fuzzy sets related to $\overline{m}_{ssov}$ are adapted depending on the average concentration of TSS. Table 2 shows fuzzy sets parameters corresponding to (28).

$$\mu_{negligible}(\overline{m}_{ssov}, k) = \begin{cases} 1, & \text{if } \overline{m}_{ssov} < a \\ \frac{b-\overline{m}_{ssov}}{b-a}, & \text{if } a \leq \overline{m}_{ssov} < b \\ 0, & \text{if } \overline{m}_{ssov} \geq b \end{cases} \quad \mu_{noticeable}(\overline{m}_{ssov}, k) = \begin{cases} 0, & \text{if } \overline{m}_{ssov} < a \\ \frac{\overline{m}_{ssov}-a}{b-a}, & \text{if } a \leq \overline{m}_{ssov} < b \\ 1, & \text{if } \overline{m}_{ssov} \geq b \end{cases} \quad (28)$$

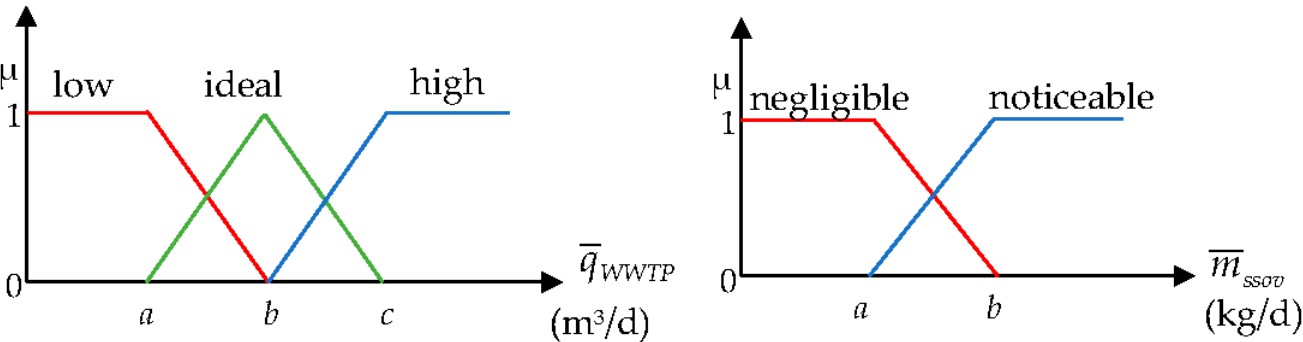

**Figure 4.** Fuzzy sets for $\overline{q}_{WWTP}$ and $\overline{m}_{ssov}$.

**Table 2.** Fuzzy set parameters.

|  | *a* | *b* | *c* |
|---|---|---|---|
| $\mu(\overline{q}_{WWTP})$ | 56,000 | 58,000 | 60,000 |
| $\mu(\overline{m}_{ssov})$ | 1000 $Tss_m(k)$ | 2000 $Tss_m(k)$ | - |

Fuzzy rules and defuzzification have been carried out following methodology of [46].

## 7. Results and Discussion

This section presents some simulation results. To reduce the computational time, a period of *Tsim* = 10 days has been taken out from a two-year data series of the benchmark [5], where the flow variability is more significant and represents a specific period of a humid season with heavy rainfall (Figure 5) with their concentrations of suspended solids (Figure 6) and ammonia (Figure 7) as representative pollutants.

To assess the methodology presented in the article, four cases have been considered. The first case (CASE 1, without control) consists of keeping all the tank output valves fully open. The second case (CASE 2, DMPC based on a cooperative game) considers two local MPCs in relation to the models shown in (15) and a negotiation based on a cooperative game shown in [45]. The third case (CASE 3, DMPC with fuzzy negotiation) corresponds to the methodology exposed in this work, and centralized MPC is CASE 4.

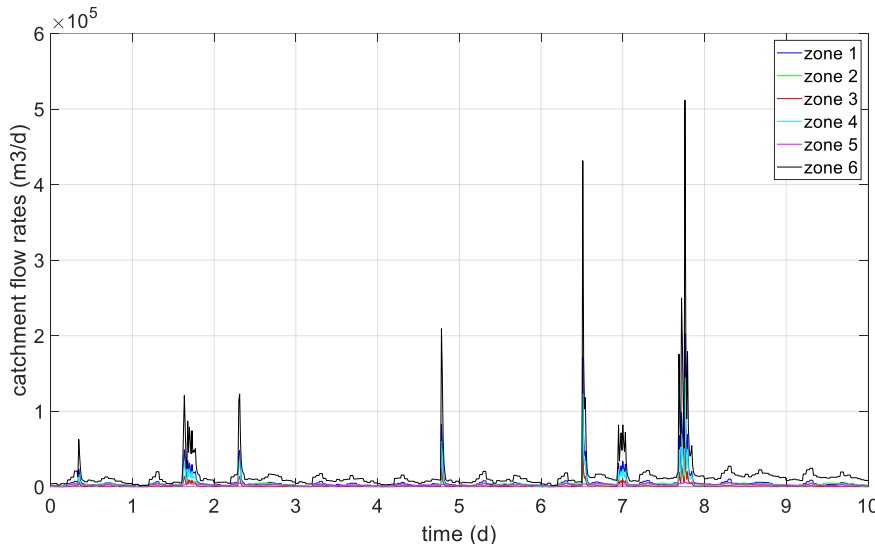

**Figure 5.** Catchment flow rates.

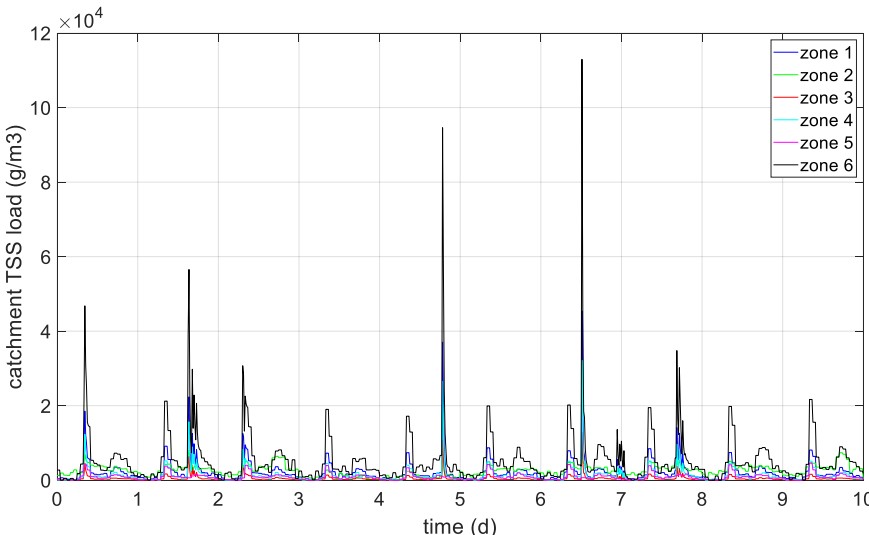

**Figure 6.** Catchment TSS load.

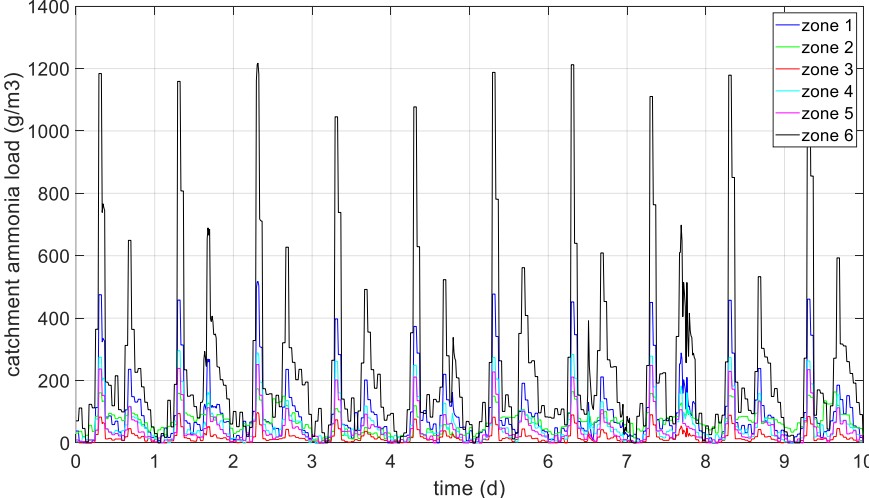

**Figure 7.** Catchment ammonia load.

In addition, for each control algorithm, two options have been considered depending on whether concentrations of TSS are included or not. The collected water comes from the urban wastewater and intensity variable rainfall events in relation to the season, which are the ones that usually cause the overflow in the deposits and at the input of the WWTP. For all cases, the prediction horizon selected is $N = 5$, and the weights in the cost function are shown in Table 3.

**Table 3.** Algorithm tuning parameters.

| Weights for (19) | Weights for (20), (21) | Weights for (22) |
|---|---|---|
| $w_i = 1, i = 1, 2, 3$ | $f_i = 10, i = 1, 2, 3, 4, 5$<br>$\alpha_i = 0, i = 1$<br>$\alpha_i = 10, i = 2, 3, 4, 5$<br>$q_{99} = 10^{-5}$<br>$q_{ii} = 10^{-12}, i = 10, 11, 12$ | $r_{ii} = 10^{-8}, i = 1, \ldots, 5$ |

The system parameters are shown in Table 4 (extracted from [5]).

**Table 4.** System parameters.

| Parameter | Units | Values |
|---|---|---|
| $A_1, \ldots, A_5$ tank areas | m$^2$ | 1188, 252, 348, 852, 2988 |
| $c_{01}, \ldots, c_{05}$ discharge coefficients | m$^{5/2}$/d | 1.89, 0.40, 0.55, 1.36, 6.12 ($\times 10^4$) |
| $h_{max1}, \ldots, h_{max5}$ tank heights | m | 5 (for all) |
| $h_{min1}, \ldots, h_{min5}$ minimum levels | m | 0 (for all) |
| $q_{max1}, \ldots, q_{max9}$ maximum flow rates at the pipes outlet | m$^3$/d | 5.99, 1.27, 3.02, 4.29, 4.29, 15.06, 4.29, 23.64, 6 ($\times 10^4$) |
| $T$ sampling time | d | 0.0021 |
| $\tau_1, \ldots, \tau_9$ link elements time constants | d | 0.0313, 0.0104, 0.0104, 0.0208, 0.0208, 0.073, 0.0208, 0.0104, 0.0104 |
| $u_{max1}, \ldots, u_{max5}$ maximum flow rates at the reservoirs outlet | m$^3$/d | 5.98, 1.27, 1.75, 4.29, 19.34 ($\times 10^4$) |
| $c_1, \ldots, c_9$ sedimentation coefficients in tanks and links | - | 0.25 (for all) |

Firstly, results without including the concentration of TSS are shown as a basis for comparison with the methodology proposed in this paper. In summary, they correspond to the methodology of [46] adapted to be applied to the sewer benchmark [5]. As you can appreciate in Figure 8, the inflow to WWTP has no significant difference for all cases (a), the same for the overflow flow rate (b) at the entrance of WWTP, corresponding with inlet flows higher than its nominal value (60,000 m$^3$/d), except for case 1. The differences between all the cases can be better appreciated in the numerical results of Table 5.

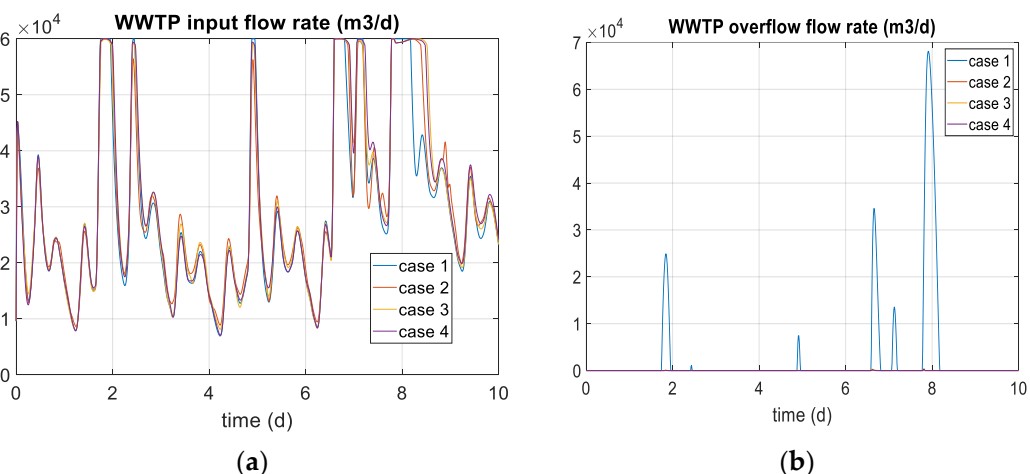

**(a)**　　　　　　　　　　　　　　　　**(b)**

**Figure 8.** WWTP input (**a**) and overflow flow rates (**b**).

Due to the water collected in the catchments having great variability, it is not easy to hold the nominal flow value at the WWTP inlet, and most of the time, it cannot reach this value. However, when any control is applied, more wastewater is retained in deposit 5, producing a better regulation of the WWTP inlet flow (Figure 8a), which is particularly noticeable at the last rainfall peak.

Figure 9 describes the dynamics of some contaminants in the WWTP provided by the benchmark that includes transport models for all pollutants (TSS and ammonia) in each case. No significant differences are observed, but, in general, it is observed that both the peaks and the valleys are more pronounced in the uncontrolled case than in the controlled cases.

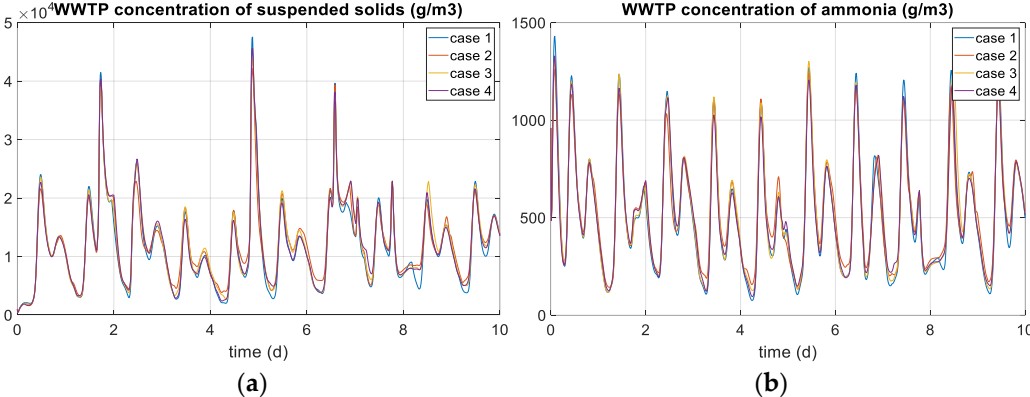

**Figure 9.** Concentration of suspended solids (**a**) and ammonia (**b**) in WWTP input flow rate.

As an example, Figure 10 shows deposit 5 level (a) and outflow set-point (b), $u_5$, in each case. The water level signal in tank 5 is shown together with the set-point calculated by the upper level of the hierarchical control system. The set-point tracking is appreciably better for cases 2, 3, and 4 than for case 1, especially at rainfall peaks. Hence, if no control is applied, most overflows are generated at the WWTP inlet because not enough water is being retained in the sewage deposits. Moreover, the controlled cases reduce the total amount of overflows at the WWTP inlet and the total overflowed water volume (Table 4).

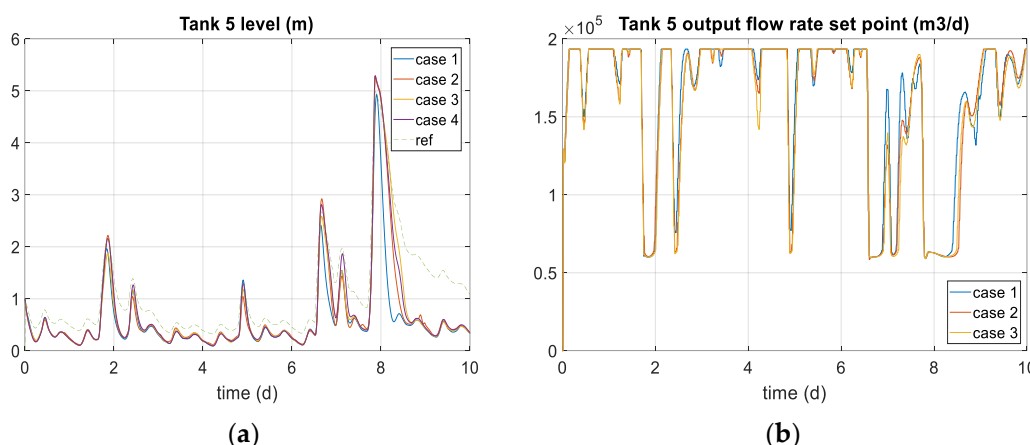

**Figure 10.** Deposit 5 level (**a**) and output flow rate reference (**b**).

The results obtained applying the methodology proposed in this work including TSS are shown in Figures 11–13. There are no significant differences comparing with previous results. Therefore, more insight is obtained from Tables 6 and 7.

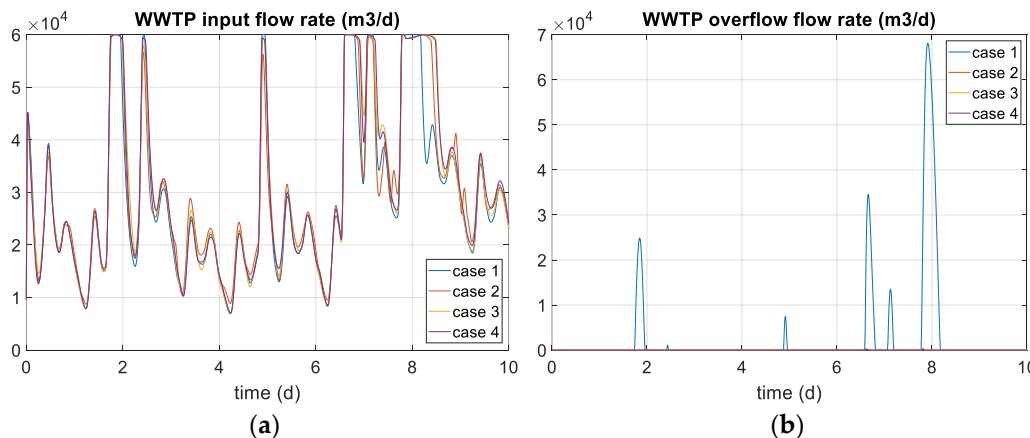

**Figure 11.** WWTP input (**a**) and overflow flow rates (**b**).

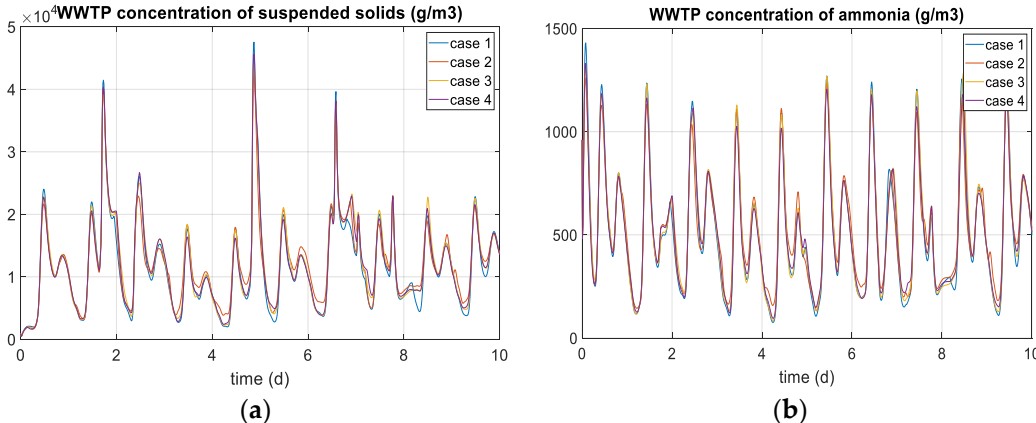

**Figure 12.** WWTP concentration of suspended solids (**a**) and ammonia (**b**).

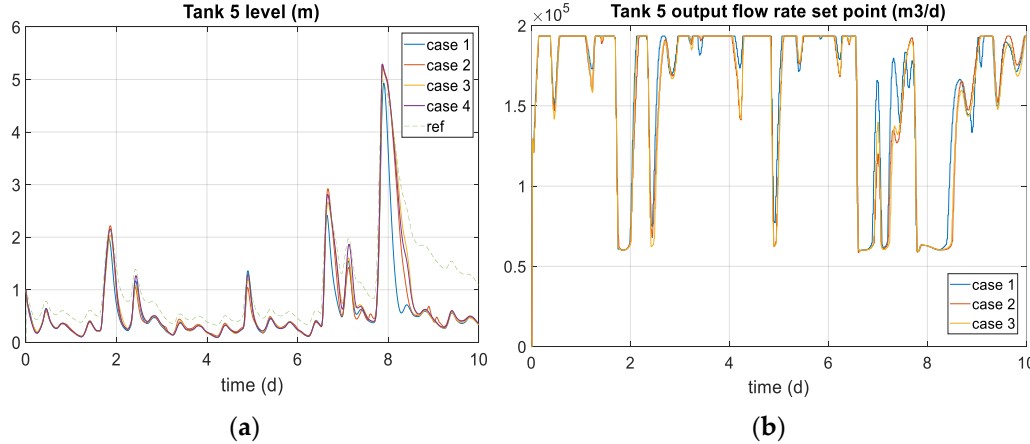

**Figure 13.** Deposit 5 level (**a**) and output flow rate set-point (**b**).

For the four cases, Tables 5 and 6 show the performance indices of Section 2.2. Case 1 (without control) presents the poorest indices, as we expected, and the centralized MPC shows the best behaviour for the average flow at the input WWTP ($Q_{WWTP}$) for the degree of usage of the treatment plant ($G_u$), for the total overflowed volume ($V_{ov}$), and for the total overflowed pollutants ($Mss_{ov}$, $NH_{ov}$, $NO_{ov}$, $PO_{ov}$). Moreover, the DMPC with fuzzy negotiation does not present a great worsening of those indices, with the advantage of using local optimization problems. Moreover, the smoothing effect of the fuzzy sets selected for negotiation is the reason why the use of fuzzy DMPC obtains the smallest $S$. The case of DMPC based on a cooperative game provides the worst performance indices of controlled

cases because the negotiation is performed in a discrete way from evaluating costs for different combinations of control actions.

**Table 5.** Numerical results for controllers without *Tss*.

| Data | No Control | DMPC with Cooperative Game | DMPC with Fuzzy Negotiation | Centralized MPC |
|---|---|---|---|---|
| $V_{ov,2}$ | 184.9266 | 312.9585 | 292.3083 | 202.1544 |
| $V_{ov,3}$ | 644.9788 | 996.5150 | 795.6183 | 681.0547 |
| $V_{ov,4}$ | $1.8077 \times 10^3$ | 6689.1 | 7968.8 | 3130.0 |
| $V_{ov,5}$ | 0 | 6577.5 | 4338.3 | 5780.0 |
| $V_{ov,WWTP}$ | $2.6362 \times 10^4$ | 18.0810 | 20.3194 | 18.0087 |
| $\boldsymbol{V_{ov}}$ | **28,999** | **14,594** | **13,415** | **9811.2** |
| $Mss_{ov,2}$ | 49.310 | 96.488 | 83.922 | 54.588 |
| $Mss_{ov,3}$ | 128.37 | 274.86 | 192.95 | 139.19 |
| $Mss_{ov,4}$ | 207.37 | 992.50 | 1131.2 | 443.44 |
| $Mss_{ov,5}$ | 0 | 841.16 | 519.96 | 712.53 |
| $Mss_{ov,WWTP}$ | 5451.6 | 6.1879 | 7.0636 | 5.9378 |
| $\boldsymbol{Mss_{ov}}$ | **5836.6** | **2211.2** | **1935.1** | **1355.7** |
| $NH_{ov,2}$ | 1.2605 | 2.4402 | 2.1236 | 1.3998 |
| $NH_{ov,3}$ | 3.2733 | 7.1891 | 4.9912 | 3.5657 |
| $NH_{ov,4}$ | 6.6447 | 33.5894 | 35.7513 | 14.9230 |
| $NH_{ov,5}$ | 0 | 27.0282 | 16.4251 | 22.6801 |
| $NH_{ov,WWTP}$ | 126.8435 | 0.1085 | 0.1033 | 0.1024 |
| $\boldsymbol{NH_{ov}}$ | **138.0221** | **70.3554** | **59.3945** | **42.6710** |
| $NO_{ov,2}$ | 0 | 0 | 0 | 0 |
| $NO_{ov,3}$ | 0 | 0 | 0 | 0 |
| $NO_{ov,4}$ | 0.0291 | 0.2452 | 0.4307 | 0.1101 |
| $NO_{ov,5}$ | 0 | 0.0991 | 0.0495 | 0.0772 |
| $NO_{ov,WWTP}$ | 0.3251 | 0.0013 | $6.3448 \times 10^{-4}$ | $6.1315 \times 10^{-4}$ |
| $\boldsymbol{NO_{ov}}$ | **0.3542** | **0.3456** | **0.4808** | **0.1879** |
| $PO_{ov,2}$ | 0.1136 | 0.2168 | 0.1916 | 0.1254 |
| $PO_{ov,3}$ | 0.3818 | 0.7977 | 0.5628 | 0.4135 |
| $PO_{ov,4}$ | 1.2249 | 7.3240 | 6.7831 | 3.1694 |
| $PO_{ov,5}$ | 0 | 5.7923 | 3.4800 | 4.8845 |
| $PO_{ov,WWTP}$ | 26.8422 | 0.0242 | 0.0230 | 0.0227 |
| $\boldsymbol{PO_{ov}}$ | **28.5626** | **14.1550** | **11.0405** | **8.6155** |
| $OQI_2$ | 23.6549 | 36.6397 | 33.1744 | 25.1296 |
| $OQI_3$ | 45.5329 | 86.6197 | 63.6188 | 48.5771 |
| $OQI_4$ | 71.5591 | 310.2451 | 344.5962 | 143.8839 |
| $OQI_5$ | 10 | 259.9951 | 163.6654 | 221.1111 |
| $OQI_{WWTP}$ | 1483.9 | 11.5668 | 11.7255 | 11.4979 |
| $\boldsymbol{OQI}$ | **1634.6** | **705.0665** | **616.7803** | **450.1995** |
| $\boldsymbol{Q_{WWTP}}$ | **28,860** | **30,021** | **30,384** | **30,556** |
| $\boldsymbol{G_u}$ | **48.1006** | **50.0351** | **50.6408** | **50.9265** |
| $S$ | - | $1.7334 \times 10^{11}$ | $6.9231 \times 10^{10}$ | $7.6086 \times 10^{10}$ |

The overflow volume in deposit 1 comes back to the sewage due to the system structure. Hence, no parameter is considered.

**Table 6.** Numerical results for TSS-included option.

| Data | No Control | DMPC with Cooperative Game | DMPC with Fuzzy Negotiation | Centralized MPC |
|---|---|---|---|---|
| $V_{ov,2}$ | 184.9266 | 313.2993 | 201.0480 | 203.8316 |
| $V_{ov,3}$ | 644.9788 | 992.7195 | 672.6625 | 666.7686 |
| $V_{ov,4}$ | $1.8077 \times 10^3$ | 6694.7 | 7765.2 | 3123.6 |
| $V_{ov,5}$ | 0 | 6682.0 | 4110.8 | 5789.3 |
| $V_{ov,WWTP}$ | $2.6362 \times 10^4$ | 19.1941 | 24.4878 | 17.9733 |
| $\boldsymbol{V_{ov}}$ | **28,999** | **14,702** | **12,774** | **9801.5** |

**Table 6.** *Cont.*

| Data | No Control | DMPC with Cooperative Game | DMPC with Fuzzy Negotiation | Centralized MPC |
|---|---|---|---|---|
| $Mss_{ov,2}$ | 49.310 | 97.191 | 54.191 | 55.108 |
| $Mss_{ov,3}$ | 128.37 | 275.65 | 143.04 | 136.24 |
| $Mss_{ov,4}$ | 207.37 | 993.03 | 1078.7 | 444.60 |
| $Mss_{ov,5}$ | 0 | 857.42 | 478.05 | 713.47 |
| $Mss_{ov,WWTP}$ | 5451.6 | 6.5113 | 8.3917 | 5.9371 |
| $\mathbf{Mss_{ov}}$ | **5836.6** | **2229.8** | **1762.4** | **1355.3** |
| $NH_{ov,2}$ | 1.2605 | 2.4591 | 1.3826 | 1.4142 |
| $NH_{ov,3}$ | 3.2733 | 7.2185 | 3.6652 | 3.4848 |
| $NH_{ov,4}$ | 6.6447 | 33.4184 | 34.3609 | 14.9650 |
| $NH_{ov,5}$ | 0 | 27.4838 | 15.1051 | 22.7104 |
| $NH_{ov,WWTP}$ | 126.8435 | 0.1183 | 0.1334 | 0.1016 |
| $\mathbf{NH_{ov}}$ | **138.0221** | **70.6981** | **59.6472** | **42.6760** |
| $NO_{ov,2}$ | 0 | 0 | 0 | 0 |
| $NO_{ov,3}$ | 0 | 0 | 0 | 0 |
| $NO_{ov,4}$ | 0.0291 | 0.2409 | 0.4353 | 0.1041 |
| $NO_{ov,5}$ | 0 | 0.0998 | 0.0507 | 0.0738 |
| $NO_{ov,WWTP}$ | 0.3251 | 0.0013 | $8.8362 \times 10^{-4}$ | $5.7748 \times 10^{-4}$ |
| $\mathbf{NO_{ov}}$ | **0.3542** | **0.3421** | **0.4869** | **0.1784** |
| $PO_{ov,2}$ | 0.1136 | 0.2182 | 0.1249 | 0.1265 |
| $PO_{ov,3}$ | 0.3818 | 0.8010 | 0.4223 | 0.4045 |
| $PO_{ov,4}$ | 1.2249 | 7.2788 | 6.2459 | 3.1790 |
| $PO_{ov,5}$ | 0 | 5.8853 | 3.1999 | 4.8879 |
| $PO_{ov,WWTP}$ | 26.8422 | 0.0263 | 0.0296 | 0.0225 |
| $\mathbf{PO_{ov}}$ | **28.5626** | **14.2096** | **10.0227** | **8.6205** |
| $OQI_2$ | 23.6549 | 36.8371 | 24.9986 | 25.2768 |
| $OQI_3$ | 45.5329 | 86.8660 | 49.6458 | 47.7420 |
| $OQI_4$ | 71.5591 | 309.8305 | 329.8800 | 144.2374 |
| $OQI_5$ | 10 | 264.6245 | 151.2962 | 221.3869 |
| $OQI_{WWTP}$ | 1483.9 | 11.6611 | 12.0823 | 11.4949 |
| $\mathbf{OQI}$ | **1634.6** | **709.8192** | **567.9028** | **450.1381** |
| $\mathbf{Q_{WWTP}}$ | **28,860** | **30,003** | **30,430** | **30,555** |
| $\mathbf{G_u}$ | **48.1006** | **50.0050** | **50.7169** | **50.9255** |
| $S$ | - | $1.7073 \times 10^{11}$ | $6.9069 \times 10^{10}$ | $7.6321 \times 10^{10}$ |

**Table 7.** Comparative improvement for controllers including TSS.

| Data | Included TSS Option | DMPC with Cooperative Game | DMPC with Fuzzy Negotiation | Centralized MPC |
|---|---|---|---|---|
| $V_{ov}$ | No | 14,594 | 13,415 | 9811.2 |
| $\mathbf{V_{ov}}$ | **Yes** | **14,702** | **12,774** | **9801.5** |
| $Mss_{ov}$ | No | 2211.2 | 1935.1 | 1355.7 |
| $\mathbf{Mss_{ov}}$ | **Yes** | **2229.8** | **1762.4** | **1355.3** |
| $NH_{ov}$ | No | 70.3554 | 59.3945 | 42.6710 |
| $\mathbf{NH_{ov}}$ | **Yes** | **70.6981** | **59.6472** | **42.6760** |
| $NO_{ov}$ | No | 0.3456 | 0.4808 | 0.1879 |
| $\mathbf{NO_{ov}}$ | **Yes** | **0.3421** | **0.4869** | **0.1784** |
| $PO_{ov}$ | No | 14.1550 | 11.0405 | 8.6155 |
| $\mathbf{PO_{ov}}$ | **Yes** | **14.2096** | **10.0227** | **8.6205** |
| $OQI$ | No | 705.0665 | 616.7803 | 450.1995 |
| $\mathbf{OQI}$ | **Yes** | **709.8192** | **567.9028** | **450.1381** |
| $Q_{WWTP}$ | No | 30,021 | 30,384 | 30,556 |
| $\mathbf{Q_{WWTP}}$ | **Yes** | **30,003** | **30,430** | **30,555** |
| $G_u$ | No | 50.0351 | 50.6408 | 50.9265 |
| $\mathbf{G_u}$ | **Yes** | **50.0050** | **50.7169** | **50.9255** |

Regarding the numerical results, the performance improves when the concentration of pollutant is considered, except in case 2, although this is not very significant.

However, the best performance improvement is in case 3 because the negotiation between agents considers the overflowed TSS mass. Table 7 shows the comparative performance improvement in each case.

In the end, to analyze how the location of the fuzzy sets affects the behavior of the DMPC algorithm, fuzzy sets have been moved, as shown in Table 8, while preserving their shape.

**Table 8.** Fuzzy sets parameters for each DMPC case.

| Case | $\mu(\overline{q}_{WWTP})$ | | | $\mu(\overline{m}_{ssov})$ | |
|------|----|----|----|----|----|
| | **a** | **b** | **c** | **a** | **b** |
| DMPC 1 | 56,000 | 58,000 | 60,000 | 1000 $Tss_m(k)$ | 2000 $Tss_m(k)$ |
| DMPC 2 | 54,000 | 56,000 | 58,000 | 1000 $Tss_m(k)$ | 2000 $Tss_m(k)$ |
| DMPC 3 | 58,000 | 60,000 | 62,000 | 1000 $Tss_m(k)$ | 2000 $Tss_m(k)$ |
| DMPC 4 | 56,000 | 58,000 | 60,000 | 0000 $Tss_m(k)$ | 1000 $Tss_m(k)$ |
| DMPC 5 | 56,000 | 58,000 | 60,000 | 2000 $Tss_m(k)$ | 3000 $Tss_m(k)$ |

The results not including the concentration of suspended solids (without TSS) are presented in Table 9, and the results including TSS are presented in Table 10.

For the first situation, the DMPC1 and DMPC2 cases show similar results, so the influence of the place of the fuzzy sets is very small. Regarding DMPC3 and DMPC2, the degree of usage of the treatment plant is slightly smaller in DMPC3 because the fuzzy set has been moved to the inlet flow constraint of 60,000 $m^3/d$. By comparing DMPC4 and DMPC3, the overflow ($V_{ov}$) is larger for DMPC3 because the fuzzy set does not consider minor overflows. In conclusion, the results have no significant differences because the fuzzy sets are very close to each other, but DMPC4 presents the best performance.

**Table 9.** Numerical results moving DMPC fuzzy sets (without including TSS).

| Data | DMPC1 | DMPC2 | DMPC3 | DMPC4 | DMPC5 |
|------|-------|-------|-------|-------|-------|
| $V_{ov,2}$ | 292.3083 | 294.4305 | 310.7161 | 261.1050 | 296.7159 |
| $V_{ov,3}$ | 795.6183 | 807.0578 | 971.8244 | 687.2200 | 835.2211 |
| $V_{ov,4}$ | 7968.8 | 7974.8 | 7721.3 | 7905.4 | 7820.3 |
| $V_{ov,5}$ | 4338.3 | 4341.4 | 4338.3 | 4338.3 | 4343.8 |
| $V_{ov,WWTP}$ | 20.3194 | 20.1841 | 21.2225 | 20.6859 | 20.1757 |
| $\boldsymbol{V_{ov}}$ | **13,415** | **13,438** | **13,982** | **13,189** | **13,316** |
| $Mss_{ov,2}$ | 83.922 | 84.702 | 93.652 | 72.960 | 85.670 |
| $Mss_{ov,3}$ | 192.95 | 196.94 | 259.77 | 155.33 | 206.23 |
| $Mss_{ov,4}$ | 1131.2 | 1133.0 | 1161.4 | 1129.1 | 1102.1 |
| $Mss_{ov,5}$ | 519.96 | 520.58 | 613.08 | 513.98 | 521.12 |
| $Mss_{ov,WWTP}$ | 7.0636 | 7.0119 | 6.9094 | 7.2022 | 6.9319 |
| $\boldsymbol{Mss_{ov}}$ | **1935.1** | **1942.2** | **2134.8** | **1878.5** | **1922.0** |
| $NH_{ov,2}$ | 2.1236 | 2.1431 | 2.3672 | 1.8504 | 2.1673 |
| $NH_{ov,3}$ | 4.9912 | 5.0960 | 6.7716 | 4.0041 | 5.3391 |
| $NH_{ov,4}$ | 35.7513 | 35.7867 | 39.8640 | 36.1368 | 34.5933 |
| $NH_{ov,5}$ | 16.4251 | 16.4406 | 19.8800 | 16.2649 | 16.4578 |
| $NH_{ov,WWTP}$ | 0.1033 | 0.1027 | 0.1165 | 0.1056 | 0.1034 |
| $\boldsymbol{NH_{ov}}$ | **59.3945** | **59.5692** | **68.9993** | **58.3617** | **58.6610** |
| $NO_{ov,2}$ | 0 | 0 | 0 | 0 | 0 |
| $NO_{ov,3}$ | 0 | 0 | 0 | 0 | 0 |
| $NO_{ov,4}$ | 0.4307 | 0.4410 | 0.3890 | 0.4154 | 0.4276 |
| $NO_{ov,5}$ | 0.0495 | 0.0505 | 0.0643 | 0.0499 | 0.0500 |
| $NO_{ov,WWTP}$ | $6.3448 \times 10^{-4}$ | $6.3597 \times 10^{-4}$ | $8.1386 \times 10^{-4}$ | $6.5225 \times 10^{-4}$ | $6.2756 \times 10^{-4}$ |
| $\boldsymbol{NO_{ov}}$ | **0.4808** | **0.4921** | **0.4541** | **0.4660** | **0.4782** |
| $PO_{ov,2}$ | 0.1916 | 0.1933 | 0.2114 | 0.1675 | 0.1953 |
| $PO_{ov,3}$ | 0.5628 | 0.5744 | 0.7573 | 0.4548 | 0.6013 |
| $PO_{ov,4}$ | 6.7831 | 6.7948 | 8.5241 | 6.8598 | 6.5964 |
| $PO_{ov,5}$ | 3.4800 | 3.4835 | 4.2674 | 3.4520 | 3.4862 |
| $PO_{ov,WWTP}$ | 0.0230 | 0.0228 | 0.0260 | 0.0235 | 0.0230 |
| $\boldsymbol{PO_{ov}}$ | **11.0405** | **11.0687** | **13.7862** | **10.9576** | **10.9022** |

**Table 9.** *Cont.*

| Data | DMPC1 | DMPC2 | DMPC3 | DMPC4 | DMPC5 |
|------|-------|-------|-------|-------|-------|
| $OQI_2$ | 33.1744 | 33.3891 | 35.8530 | 30.1599 | 33.6553 |
| $OQI_3$ | 63.6188 | 64.7335 | 82.3436 | 53.1236 | 67.3239 |
| $OQI_4$ | 344.5962 | 345.0823 | 363.1114 | 345.3249 | 335.2797 |
| $OQI_5$ | 163.6654 | 163.8365 | 192.7466 | 161.9850 | 163.9967 |
| $OQI_{WWTP}$ | 11.7255 | 11.7134 | 11.7349 | 11.7601 | 11.6996 |
| **$OQI$** | **616.7803** | **618.7548** | **685.7893** | **602.3535** | **611.9552** |
| **$Q_{WWTP}$** | **30,384** | **30,381** | **30,317** | **30,401** | **30,399** |
| **$G_u$** | **50.6408** | **50.6354** | **50.5288** | **50.6678** | **50.6643** |
| $S$ | $6.9231 \times 10^{10}$ | $6.9375 \times 10^{10}$ | $9.6167 \times 10^{10}$ | $6.9317 \times 10^{10}$ | $6.9414 \times 10^{10}$ |

For the second case, the place of the fuzzy sets affects more than before. This influence is more difficult to establish since the fuzzy sets depend at each instant on the concentration of suspended solids. However, the results have no great differences due to the fuzzy sets being very close to each other, but DMPC3 presents the best performance.

**Table 10.** Numerical results moving DMPC fuzzy sets (including TSS).

| Data | DMPC1 | DMPC2 | DMPC3 | DMPC4 | DMPC5 |
|------|-------|-------|-------|-------|-------|
| $V_{ov,2}$ | 203.5121 | 204.8541 | 288.4027 | 204.9085 | 205.5031 |
| $V_{ov,3}$ | 672.5447 | 675.6874 | 722.9298 | 673.0609 | 674.1634 |
| $V_{ov,4}$ | 7770.5 | 7701.0 | 6205.4 | 7811.8 | 7880.9 |
| $V_{ov,5}$ | 4109.7 | 4142.8 | 4873.0 | 4108.9 | 4113.2 |
| $V_{ov,WWTP}$ | 24.4633 | 24.5810 | 23.1997 | 24.4242 | 24.3320 |
| **$V_{ov}$** | **12,781** | **12,749** | **12,113** | **12,823** | **12,898** |
| $Mss_{ov,2}$ | 54.9411 | 55.2451 | 82.6870 | 55.3666 | 55.5478 |
| $Mss_{ov,3}$ | 142.9723 | 143.8091 | 170.0461 | 143.3309 | 144.106 |
| $Mss_{ov,4}$ | 1079.5 | 1073.9 | 880.6759 | 1083.5 | 1090.4 |
| $Mss_{ov,5}$ | 477.9512 | 482.4708 | 588.7328 | 477.9763 | 478.7065 |
| $Mss_{ov,WWTP}$ | 8.3843 | 8.5278 | 7.6018 | 8.3672 | 8.3379 |
| **$Mss_{ov}$** | **1763.7** | **1763.9** | **1729.7** | **1768.5** | **1777.1** |
| $NH_{ov,2}$ | 1.4013 | 1.4093 | 2.0927 | 1.4119 | 1.4165 |
| $NH_{ov,3}$ | 3.6634 | 3.6850 | 4.3927 | 3.6732 | 3.6952 |
| $NH_{ov,4}$ | 34.3839 | 34.2258 | 28.7100 | 34.5066 | 34.6885 |
| $NH_{ov,5}$ | 15.1017 | 15.2228 | 18.9341 | 15.1034 | 15.1237 |
| $NH_{ov,WWTP}$ | 0.1332 | 0.1324 | 0.1287 | 0.1330 | 0.1323 |
| **$NH_{ov}$** | **54.6835** | **54.6752** | **54.2582** | **54.8281** | **55.0561** |
| $NO_{ov,2}$ | 0 | 0 | 0 | 0 | 0 |
| $NO_{ov,3}$ | 0 | 0 | 0 | 0 | 0 |
| $NO_{ov,4}$ | 0.4302 | 0.4229 | 0.3456 | 0.4393 | 0.4273 |
| $NO_{ov,5}$ | 0.0501 | 0.0502 | 0.0642 | 0.0507 | 0.0487 |
| $NO_{ov,WWTP}$ | $8.7218 \times 10^{-4}$ | $8.4314 \times 10^{-4}$ | $8.8063 \times 10^{-4}$ | $8.8474 \times 10^{-4}$ | $8.4302 \times 10^{-4}$ |
| **$NO_{ov}$** | **0.4812** | **0.4739** | **0.4108** | **0.4909** | **0.4768** |
| $PO_{ov,2}$ | 0.1267 | 0.1274 | 0.1889 | 0.1276 | 0.1280 |
| $PO_{ov,3}$ | 0.4221 | 0.4246 | 0.4967 | 0.4231 | 0.4252 |
| $PO_{ov,4}$ | 6.2459 | 6.2178 | 5.7592 | 6.2710 | 6.2872 |
| $PO_{ov,5}$ | 3.1989 | 3.2246 | 4.0418 | 3.1996 | 3.2018 |
| $PO_{ov,WWTP}$ | 0.0294 | 0.0295 | 0.0290 | 0.0296 | 0.0294 |
| **$PO_{ov}$** | **10.0232** | **10.0238** | **10.5156** | **10.0509** | **10.0717** |
| $OQI_2$ | 25.2049 | 25.2895 | 32.8343 | 25.3219 | 25.3717 |
| $OQI_3$ | 49.6267 | 49.8594 | 57.2369 | 49.7280 | 49.9564 |
| $OQI_4$ | 330.1006 | 328.4947 | 273.1868 | 331.2860 | 333.1983 |
| $OQI_5$ | 151.2654 | 152.5351 | 185.0174 | 151.2762 | 151.4811 |
| **$OQI_{WWTP}$** | **12.0802** | **12.1066** | **11.9102** | **12.0762** | **12.0682** |
| **$OQI$** | **568.2778** | **568.2853** | **560.1855** | **569.6883** | **572.0757** |
| **$Q_{WWTP}$** | **30,430** | **30,436** | **30,474** | **30,428** | **30,423** |
| **$G_u$** | **50.7161** | **50.7267** | **50.7896** | **50.7131** | **50.7055** |
| $S$ | $6.9493 \times 10^{10}$ | $6.8258 \times 10^{10}$ | $8.5400 \times 10^{10}$ | $6.9441 \times 10^{10}$ | $6.9763 \times 10^{10}$ |

## 8. Conclusions

This work presents a DMPC with fuzzy negotiation applied to a simulated sewage network benchmark, offering good results in comparison with centralized MPC and DMPC based on a cooperative game. Naturally, centralized MPC obtains the best results, as this controller uses the entire model of the system. Nevertheless, the DMPC results have no great differences, but in this case, the control system manages simpler optimization problems and, like other distributed strategies, provides the system with a certain fault tolerance. In comparison with a DMPC based on a cooperative game, fuzzy negotiation improves the results significantly. In addition, the fuzzy negotiation includes expert knowledge of the sewer system considering fuzzy sets whose shape is real-time adapted to the process depending on the needs of the sewer system.

Moreover, only DMPC with a fuzzy negotiation algorithm considering the concentration of suspended solids improves the performance of the sewer system even more, reducing both the volume and the pollutant mass overflowed in the whole of the system, whereas DMPC based on a cooperative game does not and the MPC algorithm does not show a relevant improvement. It has been possible to verify that the improvement introduced by considering suspended solids in the MPC and DMPC algorithms is due more to the increase in the penalty for overflows and to the peculiar adaptive construction of fuzzy sets than to the optimization of the concentrations of suspended solids in the network.

Consequently, UDS control is fundamentally driven by level and flow dynamics, but the inclusion of the TSS concentration improves the industrial implementation of the controller.

**Author Contributions:** M.F. and P.V. conceived and defined the main approach; A.C. and M.F. performed the simulations; P.V. and M.F. took care of to the simulation interpretations; writing—review and editing, M.F. and A.C. All authors have read and agreed to the published version of the manuscript.

**Funding:** This research was funded by Spanish Government through the MICINN projects PID2019-105434RB-C31 and TED2021-129201B-I00, and the Samuel Solórzano Foundation through project FS/11-2021.

**Data Availability Statement:** Not applicable.

**Conflicts of Interest:** The authors affirm that they have no conflict of interest.

## Appendix A

Matrices of the model:

$$
\mathbf{A} = \begin{pmatrix}
1 & 0 & 0 & 0 & 0 & 0 & 0 & 0 & 0 & 0 & 0 & 0 & 0 & 0 & 0 & 0 \\
0 & 1 & 0 & 0 & 0 & 0 & 0 & 0 & 0 & 0 & 0 & 0 & 0 & 0 & 0 & 0 \\
0 & 0 & 1 & 0 & 0 & \frac{T}{A_3} & 0 & 0 & 0 & 0 & 0 & 0 & 0 & 0 & 0 & 0 \\
0 & 0 & 0 & 1 & 0 & 0 & 0 & 0 & 0 & 0 & 0 & 0 & 0 & 0 & 0 & 0 \\
0 & 0 & 0 & 0 & 1 & 0 & 0 & \frac{T}{A_5} & 0 & 0 & 0 & 0 & 0 & 0 & 0 & 0 \\
0 & 0 & 0 & 0 & 0 & (1-\frac{T}{\tau_3}) & 0 & 0 & 0 & 0 & 0 & 0 & 0 & 0 & 0 & 0 \\
0 & 0 & 0 & 0 & 0 & 0 & (1-\frac{T}{\tau_7}) & 0 & 0 & 0 & 0 & 0 & 0 & 0 & 0 & 0 \\
0 & 0 & 0 & 0 & 0 & 0 & \frac{T}{\tau_8} & (1-\frac{T}{\tau_8}) & 0 & 0 & 0 & 0 & 0 & 0 & 0 & 0 \\
0 & 0 & 0 & 0 & 0 & 0 & 0 & 0 & (1-\frac{T}{\tau_9}) & 0 & 0 & 0 & 0 & 0 & 0 & 0 \\
0 & 0 & 0 & 0 & 0 & 0 & 0 & 0 & 0 & (1-c_3) & 0 & 0 & c_3 & 0 & 0 & 0 \\
0 & 0 & 0 & 0 & 0 & 0 & 0 & 0 & 0 & c_4 K_{4Tss3out} & (1-c_4) & 0 & 0 & 0 & 0 & 0 \\
0 & 0 & 0 & 0 & 0 & 0 & 0 & 0 & 0 & 0 & 0 & (1-c_5) & 0 & 0 & c_5 & 0 \\
0 & 0 & 0 & 0 & 0 & 0 & 0 & 0 & 0 & 0 & 0 & 0 & (1-c_6) & 0 & 0 & 0 \\
0 & 0 & 0 & 0 & 0 & 0 & 0 & 0 & 0 & 0 & 0 & 0 & 0 & (1-c_7) & 0 & 0 \\
0 & 0 & 0 & 0 & 0 & 0 & c_8 K_{8q7} & 0 & 0 & 0 & 0 & 0 & 0 & c_8 K_{8Tss7out} & (1-c_8) & 0 \\
0 & 0 & 0 & 0 & 0 & 0 & 0 & 0 & 0 & 0 & 0 & 0 & 0 & 0 & 0 & (1-c_9)
\end{pmatrix},
$$

$$\mathbf{B}_p = \begin{pmatrix} -\frac{T}{A_1} & 0 & 0 & 0 & 0 \\ 0 & -\frac{T}{A_2} & 0 & 0 & 0 \\ 0 & 0 & -\frac{T}{A_3} & 0 & 0 \\ \frac{T}{A_4} & 0 & \frac{T}{A_4} & -\frac{T}{A_4} & 0 \\ 0 & 0 & 0 & 0 & -\frac{T}{A_5} \\ 0 & \frac{T}{\tau_3} & 0 & 0 & 0 \\ 0 & 0 & 0 & \frac{T}{\tau_7} & 0 \\ 0 & 0 & 0 & 0 & 0 \\ 0 & 0 & 0 & 0 & \frac{T}{\tau_9} \\ 0 & 0 & 0 & 0 & 0 \\ c_4 K_{4u1} & 0 & c_4 K_{4u3} & 0 & 0 \\ 0 & 0 & 0 & 0 & 0 \\ 0 & c_6 K_{6u2} & 0 & 0 & 0 \\ 0 & 0 & 0 & c_7 & 0 \\ 0 & 0 & 0 & 0 & 0 \\ 0 & 0 & 0 & 0 & c_9 \end{pmatrix},$$

$$\mathbf{B}_d = \begin{pmatrix} \frac{T}{A_1} & 0 & 0 & 0 & 0 & 0 & 0 & 0 & 0 & 0 & 0 & 0 \\ 0 & \frac{T}{A_2} & 0 & 0 & 0 & 0 & 0 & 0 & 0 & 0 & 0 & 0 \\ 0 & 0 & 0 & 0 & 0 & 0 & 0 & 0 & 0 & 0 & 0 & 0 \\ 0 & 0 & \frac{T}{A_4} & 0 & 0 & 0 & 0 & 0 & 0 & 0 & 0 & 0 \\ 0 & 0 & 0 & 0 & 0 & 0 & 0 & 0 & 0 & 0 & 0 & 0 \\ 0 & 0 & 0 & 0 & 0 & \frac{T}{\tau_3} & 0 & 0 & 0 & 0 & 0 & 0 \\ 0 & 0 & 0 & 0 & 0 & 0 & 0 & 0 & 0 & 0 & 0 & 0 \\ 0 & 0 & 0 & \frac{T}{\tau_8} & \frac{T}{\tau_8} & 0 & 0 & 0 & 0 & 0 & 0 & 0 \\ 0 & 0 & 0 & 0 & 0 & 0 & 0 & 0 & 0 & 0 & 0 & 0 \\ 0 & 0 & 0 & 0 & 0 & 0 & 0 & 0 & 0 & 0 & 0 & 0 \\ 0 & 0 & c_4 K_{4q4} & 0 & 0 & 0 & c_4 K_{4Tss1out} & 0 & c_4 K_{4Tss4} & 0 & 0 & 0 \\ 0 & 0 & 0 & 0 & 0 & 0 & 0 & 0 & 0 & 0 & c_5 & 0 \\ 0 & 0 & 0 & 0 & 0 & c_6 K_{6qr3} & 0 & c_6 K_{6Tss2out} & 0 & 0 & 0 & c_6 K_{6Tss3} \\ 0 & 0 & 0 & 0 & 0 & 0 & 0 & 0 & 0 & 0 & 0 & 0 \\ 0 & 0 & 0 & c_8 K_{8q5} & c_8 K_{8q6} & 0 & 0 & 0 & 0 & c_8 K_{8Tss5} & c_8 K_{8Tss6} & 0 \\ 0 & 0 & 0 & 0 & 0 & 0 & 0 & 0 & 0 & 0 & 0 & 0 \end{pmatrix},$$

## Appendix B

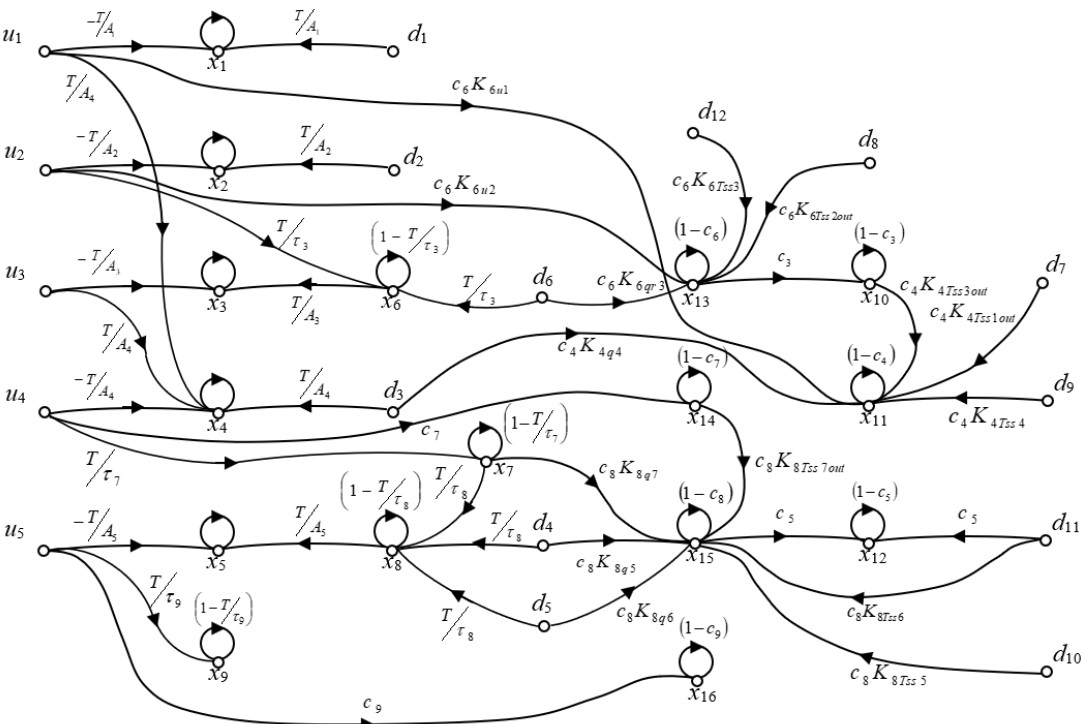

**Figure A1.** Direct graph of the model.

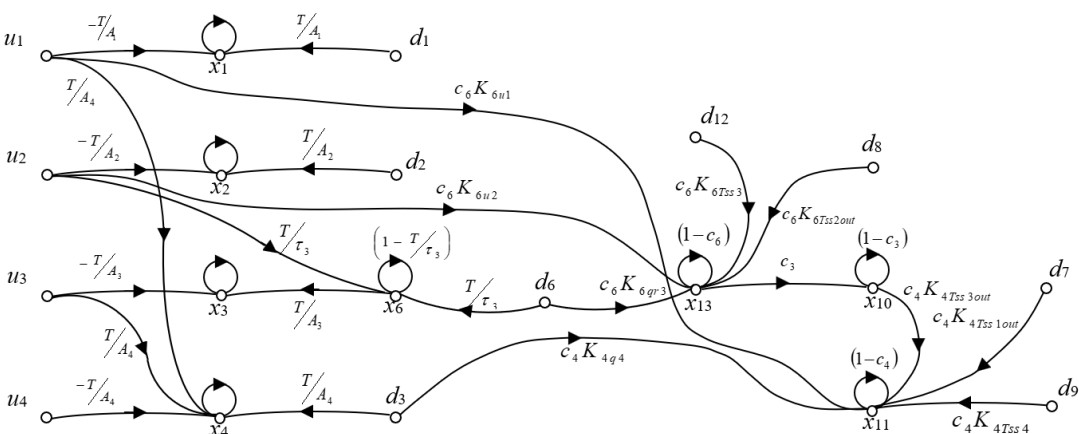

**Figure A2.** Subsystem 1.

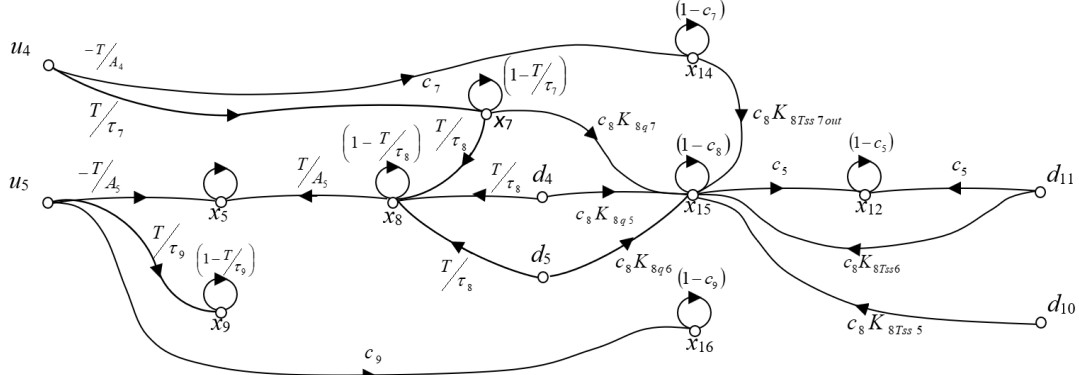

**Figure A3.** Subsystem 2.

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
