# Peer review of "Optimal Operation of a Benchmark Simulation Model for Sewer Networks Using a Qualitative Distributed Model Predictive Control Algorithm"

_processes, doi:10.3390/pr11051528_

Round 1

Reviewer 1 Report

Please find the attached report.

Author Response

Dear Reviewer 1:

Thank you very much for your instructions. In relation to them, I would like to make the following observations about the last version of the article:

- Some paragraphs of the introduction have been merged as indicated.
- Some minor citations for the article have been removed.
- A page with the nomenclature used has not been included, because we believe that to better follow the text, it is better to detail the symbology when it is used. However, when it has been considered useful to present this information jointly, tables have been used, such as for process variables (table 1) or for system parameters (table 4).
- The results shown in the figures are essentially illustrative, with the numerical results appearing in the tables being the most relevant. Even so, differences in behavior between the cases considered are not very large and further explanations are not considered necessary.

Consequently, the document has not been substantially modified and therefore no changes have been marked up.

Greetings

Reviewer 2 Report

The works is interesting, but I suppose that it needs more real situation reflection for the next. As theoretical input. The storm water modelling is characterized by the calibration and verification of the stochastic processes, so my question concern this, if is it possible to verificative this theory and how?

References 1 and 2 are not mentioned in the text.

Author Response

Dear Reviewer 2:

Thank you very much for your instructions. In relation to them, I would like to make the following observations about the latest version of the article:

- References [1] and [2] are included. It was just a mistake.
- Reference [5] describes in detail the benchmark that has been used in the work. This model represents in a realistic way, the sewer system, collecting different types of wastewater (urban, industrial) for a city of about 80,000 inhabitants. It also considers different seasons of the year and rainfall events of different intensity, using stochastic models, which are the ones that produce overflows (main problem).

The control system proposed in the article is technically viable since today all the necessary instrumentation is available for it. However, previously it would be advisable to carry out an economic feasibility study.

The document has not been substantially modified and therefore no changes have been marked.

Greetings
